# Many Minds, One Goal: Time Series Forecasting via Sub-task Specialization and Inter-agent Cooperation

Qihe Huang[†], Zhengyang Zhou[†,§,⊠], Yangze Li[†], Kuo Yang[†], Binwu Wang[†,§],
Yang Wang[†,§,⊠]

[†] University of Science and Technology of China (USTC), Hefei, China
[§] Suzhou Institute for Advanced Research, USTC, Suzhou, China
Email: {hqh,liyangze,yangkuo}@mail.ustc.edu.cn, {wbw2024, zzy0929, angyan}@ustc.edu.cn

## Abstract

Time series forecasting is a critical and complex task, characterized by diverse temporal patterns, varying statistical properties, and different prediction horizons across datasets and domains. Conventional approaches typically rely on a single, unified model architecture to handle all forecasting scenarios. However, such monolithic models struggle to generalize across dynamically evolving time series with shifting patterns. In reality, different types of time series may require distinct modeling strategies. Some benefit from homogeneous multi-scale forecasting awareness, while others rely on more complex and heterogeneous signal perception. Relying on a single model to capture all temporal diversity and structural variations leads to limited performance and poor interpretability. To address this challenge, we propose a Multi-Agent Forecasting System (MAFS) that abandons the one-size-fits-all paradigm. MAFS decomposes the forecasting task into multiple sub-tasks, each handled by a dedicated agent trained on specific temporal perspectives (e.g., different forecasting resolutions or signal characteristics). Furthermore, to achieve holistic forecasting, agents share and refine information through different communication topology, enabling cooperative reasoning across different temporal views. A lightweight voting aggregator then integrates their outputs into consistent final predictions. Extensive experiments across 11 benchmarks demonstrate that MAFS significantly outperforms traditional single-model approaches, yielding more robust and adaptable forecasts. **Code:** https://github.com/h505023992/MAFS

## 1 Introduction

Time series forecasting [79, 78, 49, 7, 2, 3, 52, 11, 13, 12, 53, 64] plays a vital role in a wide range of real-world applications, including finance, energy, healthcare, and intelligent transportation. Despite remarkable progress achieved by deep learning models such as RNNs [9, 24], CNNs [55, 22], and Transformers [60, 28, 23, 76, 25, 76], many existing approaches rely on monolithic architectures that often struggle to generalize across ever-changing temporal patterns and continuously-evolving signal characteristics inherent in time series [27, 16, 29, 51].

Meanwhile, Multi-Agent Systems (MAS) [18, 48] have emerged as a powerful paradigm for addressing complex problems through collaboration among specialized agents [75]. This framework has achieved remarkable success in a wide range of domains, including robotics [66], question answering [65], and sequential decision-making [67]. Crucially, Multi-Agent Systems allow individual agents to process information from distinct perspectives and coordinate their outputs to

---

⊠ Yang Wang and Zhengyang Zhou are corresponding authors.

achieve a common objective [74, 57]. This design promotes modularity, scalability, and robustness in solving high-dimensional and complex tasks [40, 5]. Inspired by these successes, a natural question arises: *Can Multi-Agent Systems also benefit time series forecasting, particularly in scenarios with heterogeneous temporal patterns or task decomposition requirements?*

However, directly applying multi-agent systems to time series forecasting remains non-trivial and faces three key challenges: ***(i) Task Decomposition for Specialized Modeling.*** Time series forecasting tasks are inherently unified and not easily divisible [8]. *A critical question arises: how can we decompose a global forecasting task into meaningful sub-tasks that enable each agent to develop domain-specific expertise?* Without proper decomposition, agents risk learning overlapping or redundant representations, which limits the benefits of specialization [68]. ***(ii) Limited Agent Perception.*** If an agent consistently operates within a narrow temporal scope, such as focusing only on local trends or periodic components, it may fail to capture important contextual signals needed for precise processing [74, 57]. A lack of holistic understanding can hinder the agent's ability to generalize across variable temporal conditions. ***(iii) Forecasting Collaboration Bottlenecks.*** Even when agents are well-specialized, enabling effective collaboration among them remains a substantial challenge [30, 4]. Agents must not only share information efficiently but also resolve potential conflicts in their predictions. Poor coordination can lead to inconsistent or contradictory outputs, undermining the overall forecasting accuracy [47].

To bridge the gap between monolithic forecasting models and the need for adaptive, collaborative intelligence, we propose a novel **Multi-Agent Forecasting System (MAFS)** that introduces principled mechanisms for modular time series forecasting. Fundamentally, to enable each agent to specialize in modeling a distinct temporal attribute, MAFS explores two types of subtask decomposition: (i) forecasting across multiple *homogeneous future horizons*, and (ii) predicting *heterogeneous signal features* such as frequency-domain energy, statistical moments, periodicity, and trend. This design allows each agent to learn a well-defined aspect of the temporal structure. Furthermore, to expand each agent's receptive field beyond local input views, we introduce **inter-agent communication** that allows information exchange at the representation level. This is implemented via structured message passing over predefined topologies, including *Ring*, *Star*, *Chain*, and *Fully-Connected Graphs*, enabling flexible and scalable coordination across agents. Third, to realize effective decision fusion across agents, MAFS incorporates a **two-stage voting aggregator** composed of: (i) an *Agent Confidence Estimator* that evaluates the confidence or relevance of its own prediction; and (ii) a *Global Voter* that aggregates forecasts across agents based on both their internal ratings and mutual assessments. This mechanism enables robust collaboration by assigning adaptive weights to different agents during inference. Overall, MAFS is designed to fully harness the strengths of collective intelligence while maintaining scalability and generalization across a variety of forecasting scenarios. Our contributions are summarized as follows:

- We propose **Multi-Agent Forecasting System (MAFS)**, the first general-purpose time series forecasting framework based on Multi-Agent Systems, which leverages collective intelligence to tackle complex, evolving, and heterogeneous temporal patterns.

- Within **MAFS**, we introduce two principled task decomposition strategies to enable each agent to specialize in distinct forecasting sub-tasks. Furthermore, we design an inter-agent communication module to enhance the generalization capacity of each agent through structured message passing. Finally, a two-stage voting aggregator combines self-assessments and global voting to facilitate robust and coordinated multi-agent forecasting.

- Through agent specialization and structured collaboration, MAFS demonstrates superior forecasting performance. Extensive experiments on 11 real-world datasets demonstrate that MAFS consistently outperforms competitive baselines, achieving on average a **6.35% reduction in MSE** and a **4.03% reduction in MAE** compared to its single-agent counterpart. MAFS also ranks first on **16 out of 22 metrics**, and secures a top-2 position on **20 out of 22 metrics**, spanning both MSE and MAE evaluations.

## 2 Related Work

### 2.1 Time Series Forecasting

Time series forecasting is a foundational task in machine learning that aims to predict future values from past observations [63, 36, 50, 62, 21, 70, 35, 54, 56, 43, 32, 42, 10]. Traditional statistical

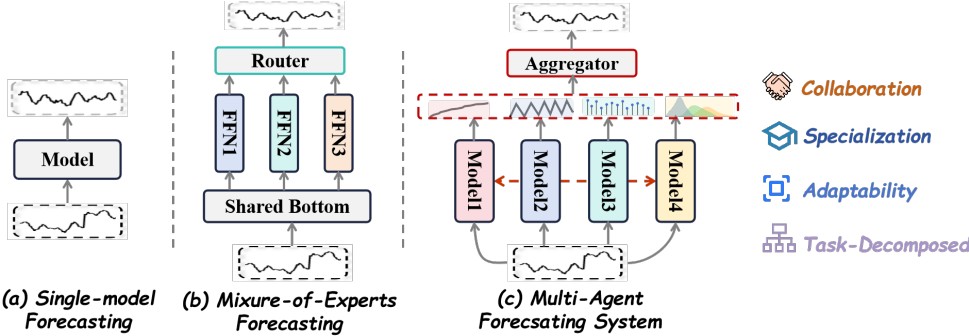

Figure 1: Illustration of different forecasting paradigms: (a) Single-model Forecasting, (b) Mixture-of-Experts Forecasting, and (c) Multi-Agent Forecasting System (MAFS). Compared to traditional paradigms, MAFS achieves better **specialization**, **collaboration**, **adaptability**, and **interpretability**.

models such as ARIMA [1] and VAR [31] are widely used for their simplicity, yet often struggle with capturing nonlinear and long-term dependencies. With the advent of deep learning, models such as RNNs [9, 24], MLPs [58, 37, 29, 34, 14], and Transformers [25, 44, 69, 77, 71, 76, 38, 33] have demonstrated remarkable improvements in modeling complex temporal dynamics. As shown in Figure 1(a), these models typically adopt monolithic architectures, where a single model is trained end-to-end to learn all patterns across the entire time series. However, such one-size-fits-all approaches often underperform in heterogeneous or nonstationary environments, where different segments of the data may exhibit distinct behaviors or require different inductive biases [16].

## 2.2 Mixture-of-Experts Forecasting

To improve modeling capacity and flexibility, Mixture-of-Experts (MoE) [45] has been applied to time series forecasting, where multiple expert models are trained to extract diverse high-level features from the input sequences. As shown in Figure 1(b), a Router mechanism is typically employed to dynamically select or weight experts for adaptive specialization across different temporal patterns. Time-MoE [46] introduces a scalable autoregressive Transformer equipped with sparse mixture-of-experts, enabling billion-scale time series pretraining with reduced inference cost and flexible forecasting horizons. From frequency aspect, MOIRAI-MoE [26] eliminates the need for human-defined frequency specialization by learning token-level expert routing within a sparse MoE framework. Despite their strengths, MoE-based forecasting models typically operate within a monolithic architecture and lack modular agent-level interpretability [20, 72].

## 2.3 Multi-Agent System

Multi-agent Systems (MAS) has emerged as a powerful paradigm for solving complex problems through the cooperation of multiple specialized agents [4]. In the context of machine learning, MAS enables agents to learn distinct competencies, share information, and coordinate actions to collectively solve tasks that are difficult for a single model [30]. Recently, there has been growing interest in leveraging large language model agents for multi-agent collaboration in open-domain reasoning [75, 65], but its application to time series forecasting remains largely unexplored. As shown in in Figure 1(c), in contrast to Mixture-of-Experts, which implicitly routes inputs through fixed expert networks without agent awareness or coordination, multi-agent systems offer explicit control, communication, and division of labor. Thus, designing an MAS forecasting system offers a promising pathway to harness specialization and collaboration for forecasting complex time series.

## 3 Problem Formulation

**Time Series Forecasting**   We consider a standard time series forecasting problem. Given a historical input sequence $\mathbf{X} = [\mathbf{x}_1, \mathbf{x}_2, .., \mathbf{x}_T] \in \mathbb{R}^{T \times M}$ with $T$ time steps and $M$ variables, the goal is to predict the future series $\mathbf{Y} = [\mathbf{x}_{T+1}, \mathbf{x}_{T+2}, .., \mathbf{x}_{T+L}] \in \mathbb{R}^{L \times M}$, where $L$ is future horizon.

**Multi-agent System for Forecasting**   To improve forecasting performance and interpretability, we introduce a multi-agent forecasting system $\mathbf{S} = \{\mathbf{Agent}_1(\cdot), \mathbf{Agent}_2(\cdot), \ldots, \mathbf{Agent}_N(\cdot)\}$

consisting of $N$ specialized forecasting agents. Each $\mathbf{Agent}_i(\cdot)$ is responsible for solving a distinct sub-task of the forecasting problem. The agents interact and exchange information via a predefined communication topology $\mathbf{G} = \{\mathbf{S}, \mathbf{A}, \mathbf{E}\}$, where $\mathbf{A} \in \{0,1\}^{N \times N}$ is a binary adjacency matrix, and $\mathbf{E} \in \mathbb{R}^{N \times N}$ denotes the edge weights. The agent outputs are aggregated by an Agent-rated Voting Aggregator $\mathrm{AVA}(\cdot)$ to produce the final forecasting result. The overall process is formulated as:

$$\hat{\mathbf{Y}} = \mathrm{AVA}(\mathrm{Comm}(\{\mathbf{Agent}_1(\mathbf{X}), \mathbf{Agent}_2(\mathbf{X}), \dots, \mathbf{Agent}_N(\mathbf{X}); \mathbf{G})) \tag{1}$$

where $\mathrm{Comm}(\cdot; \mathbf{G})$ represents the communication mechanism. Let $\Theta_A = \{\theta_1, \dots, \theta_N\}$ denote the parameters of all agents, $\Theta_T$ the parameters of communication weight (i.e., $\mathbf{E}$), and $\Theta_P$ the parameters of the Agent-rated Voting Aggregator. The joint training objective is to minimize the expected mean squared error (MSE) between the predicted and the truth:

$$\min_{\Theta_A, \Theta_T, \Theta_P} \mathbb{E}_{(\mathbf{X},\mathbf{Y}) \sim \mathcal{D}} \left[ \|\mathbf{Y} - \hat{\mathbf{Y}}\|_2^2 \right], \tag{2}$$

where $\mathcal{D}$ denotes the empirical data distribution.

## 4 Methodology

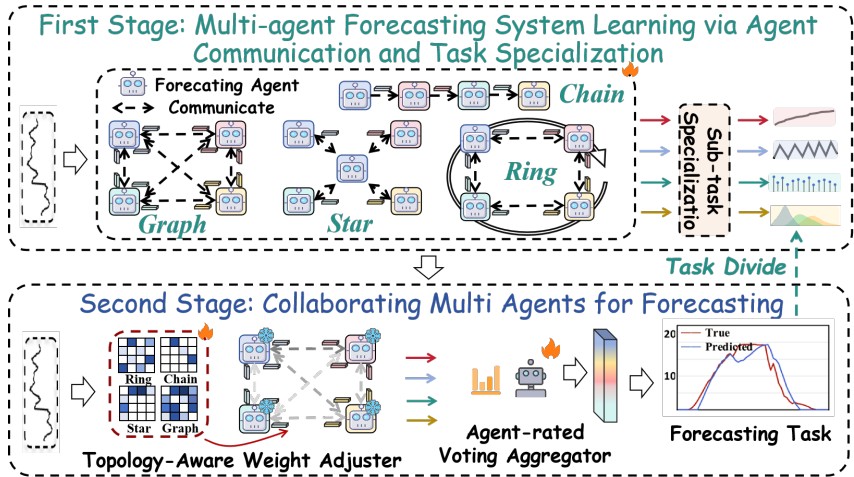

Figure 2: Overview of the two-stage learning in Multi-Agent Forecasting System.

As illustrated in Figure 2, the Multi-Agent Forecasting System (MAFS) is structured around a two-stage learning paradigm. **Stage 1: Specialization Pretraining.** Each agent is assigned a distinct sub-task (e.g., different signal characteristics) and exchanges hidden states at every layer using a fixed communication graph with uniform weights. Only agent-specific parameters $\Theta_A$ are optimized in this stage, enabling each agent to specialize independently. **Stage 2: Collaborative Forecasting.** Agent parameters are frozen. Edge weights in communication is now learnable by Topology-aware Weight Adjuster with $\Theta_T$. An Agent-rated Voting Aggregator (AVA) with parameters $\Theta_P$ aggregates the communicated features to produce the final prediction.

### 4.1 Forecasting Agent Architecture

In the Multi-Agent Forecasting System (MAFS), as shown in Figure 3, each forecasting agent adopts a pluggable encoder-based time series model as its backbone, enabling flexible integration with various architectures. Given an input sequence $\mathbf{X} \in \mathbb{R}^{T \times M}$, the encoding process is formulated as:

$$\mathbf{H}^{(0)} = \mathrm{Embed}(\mathbf{X}^\top), \quad \mathbf{H}^{(l+1)} = \mathrm{EncoderLayer}(\mathbf{H}^{(l)}), \quad \hat{\mathbf{O}} = \mathrm{Head}(\mathbf{H}^{(L)}) \tag{3}$$

Here, $\mathbf{H}^{(l)}$ represents the hidden state at the $l$-th encoder layer. The $\mathrm{EncoderLayer}(\cdot)$ can follow any standard design, making MAFS extensible to a wide range of forecasting backbones. $\hat{\mathbf{O}}$ is the agent output of specialized forecasting sub-task.

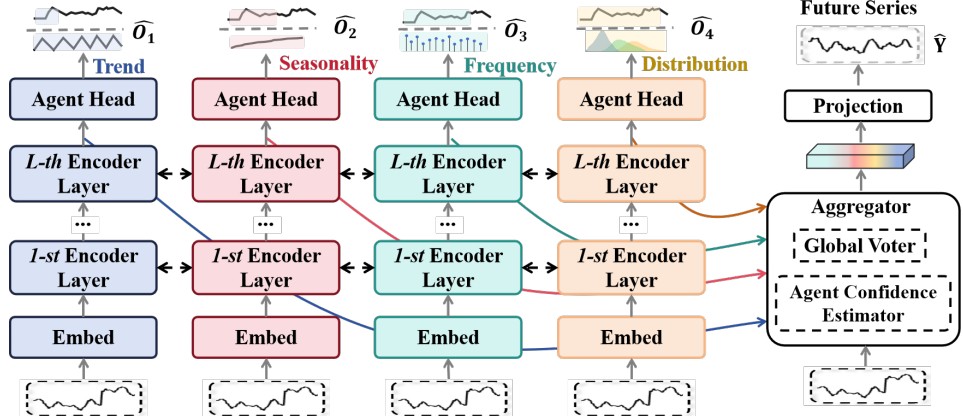

Figure 3: The overall architecture of MAFS. Each forecasting agent employs a pluggable encoder-based time series model, and their representation are aggregated to produce the final forecast.

## 4.2 Agent Communication

Each agent is assigned with a dedicated forecasting sub-task. While such specialization facilitates inductive bias and modeling diversity, it inevitably leads to data bias. To mitigate this limitation and enable agents to form coherent and globally consistent predictions, we introduce an explicit communication mechanism. This communication can be formally expressed as, $\{\mathbf{HC}_1^l, \mathbf{HC}_2^l, ..., \mathbf{HC}_N^l = \text{Comm}(\mathbf{H}_1^l, \mathbf{H}_2^l, ..., \mathbf{H}_N^l; \mathbf{G}); \mathbf{H}_i^{l+1} = \text{EncoderLayer}_i(\mathbf{HC}_i^l)\}$. Here, $\mathbf{H}_i^l$ denotes the output of the $l$-th encoder layer of agent $i$, and $\mathbf{HC}_i^l$ represents the updated representation after communication. The communication function $\text{Comm}(\cdot)$ is instantiated as a graph convolution operation, enabling structured message passing across agents connected via $\mathbf{G}$. Specifically, we implement the communication via a Graph Convolutional Network [19],

$$\mathbf{HC}_i^l = \sigma\left(\sum_{j \in \mathcal{N}(i)} \mathbf{A}_{ij} \cdot \mathbf{W} \cdot \mathbf{H}_j^l\right) \tag{4}$$

where $\mathcal{N}(i)$ is the set of connected agents of $\mathbf{Agent}_i$, $\mathbf{A}_{ij}$ is the normalized adjacency matrix indicating communication strength between agents $i$ and $j$, $\mathbf{W} \in \mathbb{R}^{d_{\text{model}} \times d_{\text{model}}}$ is a learnable projection matrix, and $\sigma(\cdot)$ is a non-linear activation function. This formulation enables each agent to refine its hidden representation by aggregating contextual information from others, thereby overcoming the limitations of isolated modeling and improving coordination among agents.

## 4.3 Agent Specialization via Sub-task Decomposition

To promote diversity and specialization among agents, we design two sub-task decomposition strategies: **(1) Multi-scale temporal forecasting (homogeneous setting):** Each agent focuses on predicting a different portion of the future horizon at increasing lengths, e.g., the $\mathbf{Agent}_i$ predicts the first $\frac{i}{N} \times L$ steps. This encourages specialization across temporal scales, from short-term to long-term forecasting. **(2) Multi-aspect signal forecasting (heterogeneous setting):** Agents are assigned complementary signal analysis tasks, including trend extraction, seasonality modeling, spectral energy estimation, and statistical summary prediction. Each sub-task targets a distinct property of the future signal, fostering diverse and orthogonal representations. These decompositions enable the agent ensemble to capture rich temporal dynamics from multiple perspectives, improving overall forecasting accuracy. More details are availavle at Appendix A.

## 4.4 Topology-aware Weight Adjuster

In the second stage of training, we freeze the parameters of each forecasting agent and make the inter-agent communication weights learnable. Specifically, the fixed edge weight matrix $\mathbf{E}$ is replaced by a parameterized matrix $\mathbf{E}_{\Theta_D} \in \mathbb{R}^{N \times N}$, where each entry denotes the learnable communication strength between a pair of agents.

To retain the prior topology, we keep a fixed binary adjacency mask $\mathbf{A} \in {0, 1}^{N \times N}$, which encodes the initial traffic-aware structure. The learnable edge weights are modulated by this mask to obtain the soft communication graph, $\mathbf{A}' = \sigma(\mathbf{E}_{\Theta_D}) \odot \mathbf{A}$, where $\sigma(\cdot)$ is the sigmoid function and $\odot$ denotes element-wise multiplication. To ensure symmetric communication [19], we construct the normalized adjacency matrix as,

$$\mathbf{A}_{\text{norm}} = \mathbf{D}^{-1/2}(\hat{\mathbf{A}})\mathbf{D}^{-1/2}, \quad \text{where} \quad \hat{\mathbf{A}} = \frac{1}{2}(\mathbf{A}' + \mathbf{A}'^{\top}) + \mathbf{I}, \quad \mathbf{D} = \text{diag}\left(\sum_j \hat{A}_{ij}\right). \quad (5)$$

This topology-aware normalization allows information exchange to be dynamically adjusted while preserving structural priors. Crucially, $\mathbf{E}_{\Theta_D}$ is optimized jointly with the forecasting objective via backpropagation, enabling the system to learn an adaptive, task-specific communication topology that enhances coordination among agents and improves overall prediction performance.

### 4.5 Agent-rated Voting Aggregator

During forecasting phase, we aggregate the final embedding from all agents to complete forecasting. To dynamically make decision-making process adaptive to each agent, we design a two-stage voting aggregator (AVA), which include an **Agent Confidence Estimator** and a **Global Voter**.

**Agent Confidence Estimator** Let $\{\mathbf{H}_1^L, \mathbf{H}_2^L, \ldots, \mathbf{H}_N^L\}$ denote the final output representations of all forecasting agents. The Agent Confidence Estimator evaluates the confidence of each agent through a self-assessment mechanism. Each $\mathbf{H}_i^L$ is paired with a shared contextual embedding $\mathbf{C}_i$, and passed through a learnable gating network to produce an element-wise gate coefficient:

$$\alpha_i = \sigma(\text{Gate}([\mathbf{C}_1, \mathbf{C}_2, \ldots, \mathbf{C}_N])) \quad (6)$$

The final gated representation is computed as:

$$\tilde{\mathbf{H}}_i = \alpha_i \odot \mathbf{H}_i^L + (1 - \alpha_i) \odot \mathbf{C}_i \quad (7)$$

Here, $\odot$ denotes element-wise multiplication, $\sigma(\cdot)$ is the sigmoid activation function, and Gate$(\cdot)$ is a shared learnable network. This formulation enables each agent to adjust its output by blending its own prediction with the global context, based on estimated confidence.

**Global Voter** To further integrate the confidence-adjusted outputs from all agents, we introduce a Global Voter that computes agent-wise collaboration weights $\mathbf{CW} \in \mathbb{R}^N$ based on the encoded input $\mathbf{X}_{\text{enc}}$ with MLP. These weights indicate the relative importance or contribution of each agent to final prediction. Each weight in $\mathbf{CW}$ is then broadcast to match the shape of its corresponding agent representation and used to perform a weighted sum over $\{\tilde{\mathbf{H}}_1^L, \tilde{\mathbf{H}}_2^L, \ldots, \tilde{\mathbf{H}}_N^L\}$. This results in a unified latent representation $\mathbf{Z}$, which captures the aggregated knowledge across all agents.

Finally, $\mathbf{Z}$ is projected through a linear layer to produce the final multivariate time series forecast $\hat{\mathbf{Y}} \in \mathbb{R}^{M \times L}$, where $L$ denotes the forecast horizon.

## 5 Experiments

### 5.1 Experimental Setups

**Datasets** We evaluate our model on 11 real-world datasets covering different domains. In Electricity domain, we use ETTh1, ETTh2, ETTm1, and ETTm2 [77, 60]. The Environment domain includes Weather [60], PM2.5 [59], AQShunyi and AQWan [41] . The Nature domain consists of CzeLan, ZafNoo [41] as well as Temp [59]. For the ETT datasets, we adopt a 6:2:2 train/validation/test split, and a 7:1:2 split for the remaining datasets.

**Implementation Details** MAFS explores four distinct communication topologies for the agent interaction graph $\mathbf{G}$, including ring, star, chain, and fully-connected structures. Each forecasting agent is implemented using iTransformer architecture [28]. MAFS is trained in two stages: the first stage independently optimizes each agent with a learning rate of 1e-3 for 10 epochs; the second stage freezes agent parameters and finetunes only the topology-aware weight adjuster and agent-rated voting aggregator for another 10 epochs with a reduced learning rate of 1e-4. We set a hidden dimension of 128, and configure the encoder with 2 layer for all datasets. The number of agents

Table 1: Comparison of long-term time series forecasting methods on 11 datasets using MSE and MAE (lower is better). Best results are marked in red ; second-best results are underlined in blue .

| Methods / Datasets | MAFS (Ours) MSE | MAE | iTransformer [2024] MSE | MAE | TimeMixer [2024] MSE | MAE | PatchTST [2024] MSE | MAE | Crossformer [2023] MSE | MAE | TiDE [2024] MSE | MAE | TimesNet [2023] MSE | MAE | DLinear [2023] MSE | MAE | Autoformer [2021] MSE | MAE | Informer [2021] MSE | MAE |
|---|---|---|---|---|---|---|---|---|---|---|---|---|---|---|---|---|---|---|---|---|
| ETTh1 | **0.433** | **0.437** | 0.467 | 0.466 | 0.447 | 0.44 | 0.469 | 0.454 | 0.529 | 0.522 | 0.541 | 0.507 | 0.458 | 0.45 | 0.456 | 0.452 | 0.496 | 0.487 | 1.04 | 0.795 |
| ETTh2 | **0.356** | **0.394** | 0.386 | 0.415 | 0.364 | 0.395 | 0.387 | 0.407 | 0.942 | 0.684 | 0.611 | 0.55 | 0.414 | 0.427 | 0.559 | 0.515 | 0.45 | 0.459 | 4.431 | 1.729 |
| ETTm1 | **0.366** | **0.388** | 0.383 | 0.403 | 0.381 | 0.395 | 0.387 | 0.4 | 0.513 | 0.496 | 0.419 | 0.419 | 0.4 | 0.406 | 0.403 | 0.407 | 0.588 | 0.517 | 0.961 | 0.734 |
| ETTm2 | **0.265** | **0.321** | 0.29 | 0.339 | 0.275 | 0.323 | 0.281 | 0.326 | 0.757 | 0.61 | 0.358 | 0.404 | 0.291 | 0.333 | 0.35 | 0.401 | 0.327 | 0.371 | 1.41 | 0.81 |
| Weather | **0.233** | **0.267** | 0.243 | 0.276 | 0.24 | 0.271 | 0.259 | 0.281 | 0.259 | 0.315 | 0.271 | 0.32 | 0.259 | 0.287 | 0.265 | 0.317 | 0.338 | 0.382 | 0.634 | 0.548 |
| AQShunyi | 0.701 | 0.509 | 0.723 | 0.515 | 0.719 | 0.529 | 0.705 | 0.509 | **0.694** | **0.504** | 0.778 | 0.554 | 0.726 | 0.516 | 0.706 | 0.522 | 0.764 | 0.541 | 0.782 | 0.545 |
| AQWan | 0.802 | 0.503 | 0.817 | 0.507 | 0.828 | 0.499 | 0.812 | 0.499 | **0.786** | **0.49** | 0.856 | 0.536 | 0.813 | 0.5 | 0.818 | 0.512 | 0.84 | 0.525 | 0.866 | 0.525 |
| CzeLan | **0.222** | **0.271** | 0.232 | 0.28 | 0.228 | 0.28 | 0.227 | 0.29 | 0.956 | 0.576 | 0.237 | 0.303 | 0.224 | 0.285 | 0.284 | 0.342 | 0.307 | 0.355 | 0.316 | 0.355 |
| ZafNoo | 0.52 | 0.451 | 0.541 | 0.468 | 0.538 | 0.44 | 0.511 | 0.465 | **0.494** | 0.455 | 0.569 | 0.498 | 0.537 | 0.465 | 0.496 | 0.451 | 0.725 | 0.599 | 0.744 | 0.602 |
| PM2.5 | **0.398** | **0.414** | 0.421 | 0.421 | 0.415 | 0.436 | 0.46 | 0.479 | 0.456 | 0.472 | 0.481 | 0.497 | 0.473 | 0.492 | 0.453 | 0.477 | 0.515 | 0.524 | 0.539 | 0.561 |
| Temp | **0.14** | **0.288** | 0.173 | 0.321 | 0.146 | 0.304 | 0.147 | 0.306 | 0.206 | 0.423 | 0.164 | 0.338 | 0.208 | 0.428 | 0.159 | 0.327 | 0.244 | 0.5 | 0.238 | 0.492 |

$N$ is selected from the range $\{4, 8, 12, 16, 20, 24\}$ to investigate the impact of varying agent scale. All experiments adopt a symmetric prediction setting, where the input sequence length equals the forecasting horizon [76]. More Details are available at Appendix C.

**Baselines** The proposed method is evaluated against a range of representative baselines, which can be categorized by model architecture. Transformer-based models include Informer [77], Autoformer [60], Crossformer [76], PatchTST [38], and iTransformer [28]. Linear-based models such as TimeMixer [58], TiDE [6] and DLinear [73] focus on efficient feature extraction and forecsating. Periodicity-based models, such as TimesNet [61], enhance forecasting performance by modeling multi-period temporal patterns.

## 5.2 Main Results

The experimental results are presented in Table 1, where we comprehensively evaluate the proposed MAFS against a range of SOTA models on long-term time series forecasting benchmarks. Experimental results consistently demonstrate the superior performance and robustness of MAFS across diverse datasets and prediction horizons. Specifically, MAFS achieves an average improvement of 3.24% in MSE and 1.71% in MAE over the recent SOTA model TimeMixer, confirming the effectiveness of our multi-agent collaborative framework. Furthermore, although MAFS adopts the iTransformer backbone, which is not the most competitive SOTA model in terms of forecasting accuracy, our multi-agent framework effectively overcomes this limitation. By leveraging agent specialization and structured inter-agent collaboration, MAFS achieves an average improvement of 6.35% in MSE and 4.03% in MAE compared to iTransformer, successfully reaching state-of-the-art performance across multiple benchmarks. In addition, MAFS ranks first on 16 out of 22 evaluation metrics and secures a top-2 ranking in 20 out of 22 metrics (covering both MSE and MAE across 11 datasets), showcasing its strong generalization capability across various application domains. In summary, MAFS breaks the limitations of monolithic forecasting models by introducing a flexible and adaptive multi-agent collaboration mechanism, delivering more accurate and reliable predictions across diverse and challenging time series forecasting scenarios. Full results are available at Appendix G.

## 5.3 Ablation Analysis

To evaluate the effectiveness of each key component in our proposed MAFS framework, we conduct a series of ablation studies. The experimental results are presented in Table 2. We evaluate the removal of three critical modules, **1) w/o Comm**: Disables inter-agent communication by removing the shared semantic vector $h_c$, preventing collaborative reasoning. **2) w/o AVA**: Removes the Agent-Rated Voting Aggregator, directly averaging agent embeddings without adaptive gating and collaboration weights. **3) w/o STS**: Disables Sub-task Specialization, assigning identical forecasting tasks to all agents instead of specialized sub-tasks.

**Main results.** (1) w/o Comm: Without communication, MSE and MAE increase by 4.18% and 2.62%, highlighting its role in enabling agents to share complementary temporal information. (2) Removing AVA leads to a significant MSE and MAE increase of 7.24% and 5.70%, confirming its importance in adaptively integrating agent outputs for accurate forecasting. (3) Without sub-task

Table 2: Ablation Study on the Contribution of Communication, Aggregation, and Specialization Modules in MAFS. **Bold** values indicate the best performance.

| Variant | Metric | ETTh1 | ETTh2 | ETTm1 | ETTm2 | Weather | AQShunyi | AQWan | CzeLan | ZafNoo | PM2.5 | Temp |
|---------|--------|-------|-------|-------|-------|---------|----------|-------|--------|--------|-------|------|
| MAFS | MSE | **0.433** | **0.356** | **0.366** | **0.265** | **0.233** | **0.701** | **0.802** | **0.222** | **0.520** | **0.398** | **0.140** |
| | MAE | **0.437** | **0.394** | **0.388** | **0.321** | **0.267** | **0.509** | **0.503** | **0.271** | **0.451** | **0.414** | **0.288** |
| w/o Comm | MSE | 0.445 | 0.364 | 0.375 | 0.277 | 0.238 | 0.719 | 0.824 | 0.228 | 0.529 | 0.439 | 0.167 |
| | MAE | 0.446 | 0.402 | 0.398 | 0.335 | 0.273 | 0.517 | 0.513 | 0.279 | 0.456 | 0.427 | 0.312 |
| w/o AVA | MSE | 0.454 | 0.373 | 0.379 | 0.288 | 0.265 | 0.722 | 0.845 | 0.253 | 0.536 | 0.421 | 0.158 |
| | MAE | 0.456 | 0.409 | 0.403 | 0.335 | 0.293 | 0.520 | 0.532 | 0.292 | 0.487 | 0.426 | 0.301 |
| w/o STS | MSE | 0.461 | 0.384 | 0.374 | 0.286 | 0.244 | 0.726 | 0.823 | 0.243 | 0.546 | 0.417 | 0.171 |
| | MAE | 0.462 | 0.412 | 0.397 | 0.332 | 0.275 | 0.517 | 0.512 | 0.285 | 0.475 | 0.415 | 0.317 |

specialization, MSE and MAE rise by 6.97% and 3.93%, demonstrating that specialized agents improve representation and capture diverse signal characteristics.

## 5.4 Evaluating the Advantage of MAFS over Single Models

As shown in Figure 4, MAFS consistently outperforms the single-agent model across 11 datasets, achieving an average improvement of 6.35% in MSE and 4.03% in MAE, demonstrating the effectiveness of collaborative forecasting. The most significant improvement occurs on the Temp dataset, with a 19.08% reduction in MSE and 10.28% in MAE. This gain likely results from strong temporal patterns and variable dependencies in temperature data, which are better captured by specialized agents and collaborative reasoning. Overall, these results confirm that MAFS effectively realizes collective intelligence, enabling specialized agents to achieve superior forecasting accuracy, and highlighting the advantages of a multi-agent framework over monolithic models.

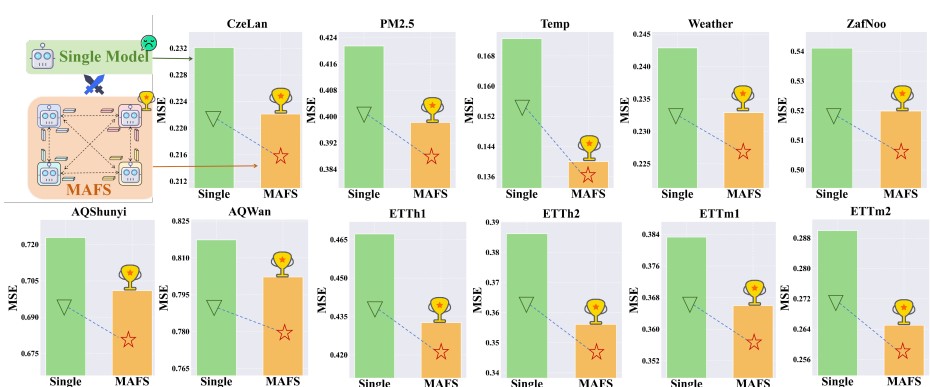

Figure 4: Performance comparison between single forecasting model and MAS forecasting system

## 5.5 Analysis of MAFS Scaling and Communication Structures

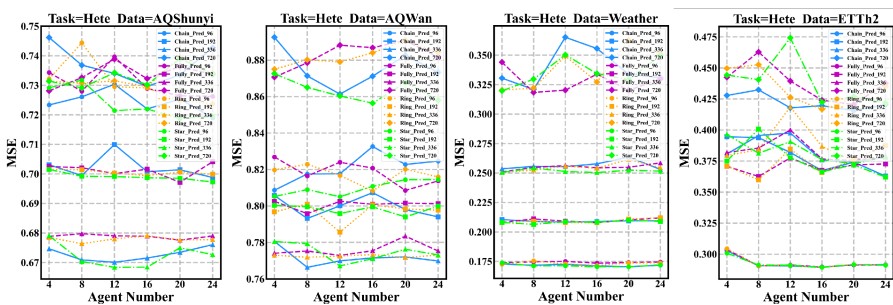

Figure 5: Performance comparison under varying agent numbers and communication structures

The results are shown in Figure 5. (1) Agent Scaling: Increasing the number of agents generally improves performance by enhancing modeling diversity and specialization. However, beyond 16 agents, improvements become marginal, indicating a saturation point. Minor performance fluctuations are trivial and likely caused by random initialization or local optima. Full results are available at Appendix E. (2) Communication Structures: Among chain, ring, fully-connected, and star topologies, the star structure consistently yields better and more stable results. Its centralized design effectively integrates global information, reducing noise amplification and ensuring critical trends are shared across agents. In conclusion, using 16 agents with a star communication structure offers the best trade-off between performance and complexity for time series forecasting.

## 5.6 Impact of Sub-task Division on Agent Specialization

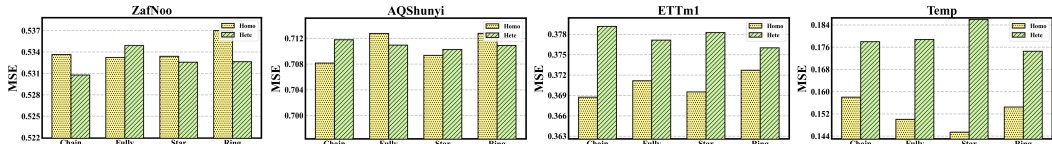

Figure 6: Comparison of agent specialization under different sub-task division strategies

Figure 6 shows the average forecasting accuracy under different sub-task divisions across prediction horizons. First, in most cases, the performance difference between homogeneous and heterogeneous task divisions is minor. For example, the performance gap remains within 1% between ZafNoo and AQShunyi, suggesting that task division strategies have limited influence under such data conditions. Second, for datasets with smaller variance and more stable patterns (e.g., ETTm1 and Temp), homogeneous task division significantly outperforms heterogeneous division. This indicates that consistent modeling strategies are better suited for stable datasets without introducing unnecessary task diversity. In summary, the choice of sub-task division should consider the characteristics of the target dataset, with homogeneous division preferred for stable data and heterogeneous division applicable when greater diversity is required. Full results are available at Appendix D.

## 5.7 Case Study

As shown in Figure 7, we take ETTm2 to illustrate effectiveness of MAFS through sub-task evaluation, voting scores of different agents, and the learned communication weights. (1) Each agent can successfully specialize and perform well on its assigned sub-task. (2) The voting results show that all agents actively contribute to the final decision, avoiding issues such as agent collapse or over-reliance on a single expert.

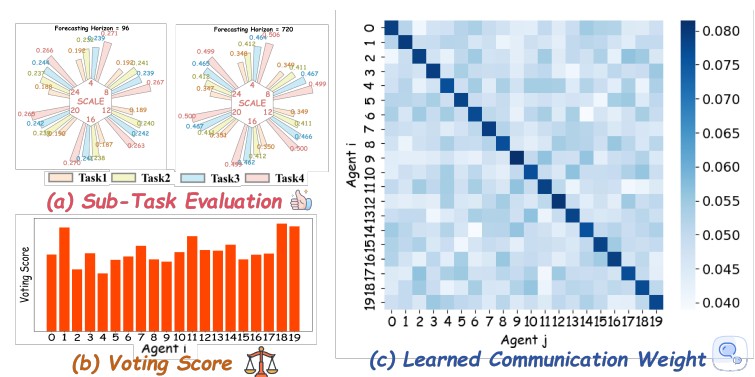

Figure 7: Case study of MAFS on ETTm2.

(3) In the second stage, the learned adaptive communication weights reflect task-aware information exchange, optimizing collaboration between agents under a fixed topology.

## 6 Conclusion

In this work, we propose MAFS, a novel multi-agent forecasting system that introduces collective intelligence into time series forecasting through principled task decomposition, structured inter-agent communication, and a two-stage voting aggregator. Extensive experiments on 11 real-world datasets demonstrate that MAFS consistently outperforms state-of-the-art baselines, achieving significant

improvements in both accuracy and robustness. These results highlight the effectiveness of collaborative forecasting and offer a new perspective for addressing the challenges of complex and evolving temporal patterns. This work can open new directions for modular, adaptive, and interpretable time series forecasting.

# 7 Acknowledgement

This paper is partially supported by the National Natural Science Foundation of China (No.62502488, No.12227901), Natural Science Foundation of Jiangsu Province (BK20240460), the grant from State Key Laboratory of Resources and Environmental Information System. The AI-driven experiments, simulations and model training were performed on the robotic AI-Scientist platform of Chinese Academy of Sciences.

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

# Appendix

## A    Details of Forecasting Sub-task Decomposition

**Homogeneous sub-task Decomposition: Multi-Scale Temporal Forecasting.** Under the homogeneous setting, each agent is tasked with predicting future values at a distinct temporal resolution. Specifically, the $i$-th agent forecasts a future subsequence of length $(L/K) \cdot i$, where $L$ is the full prediction horizon and $K$ is the number of agents. Formally, the prediction from agent $i$ is denoted as

$$\hat{\mathbf{O}}_i = \mathbf{Agent}_i(\mathbf{X}), \quad \hat{\mathbf{O}}_i \in \mathbb{R}^{(L/K) \cdot i \times M},$$

where $\mathbf{X} \in \mathbb{R}^{T \times M}$ is the multivariate historical sequence of length $T$ and $M$ is the number of variables. This design ensures that each agent captures forecasting dynamics at a specific temporal scale, ranging from short-term fluctuations to long-term evolution, thereby forming a multi-resolution ensemble.

**Heterogeneous sub-task Decomposition: Multi-Aspect Signal Forecasting.** To capture different latent components of the future signal, we further design a heterogeneous sub-task scheme, where each agent is specialized on a predefined signal property. This enables orthogonal learning objectives across agents, contributing complementary views to the ensemble. We define four types of forecasting sub-tasks in this setting:

We first define the future sequence as $\mathbf{Y} \in \mathbb{R}^{L \times M}$, where $L$ is the prediction horizon. Let each agent process $\mathbf{Y}$ to extract distinct forecasting targets.

(1) Trend forecasting. We estimate the low-frequency trend component by applying temporal average pooling over a padded version of $\mathbf{Y}$:

$$\mathbf{T} = \mathbf{AvgPool}(\mathbf{Pad}(\mathbf{Y})) \in \mathbb{R}^{T \times M},$$

where $\mathbf{Pad}(\cdot)$ aligns the length of $\mathbf{Y}$ with the pooling window and $\mathbf{AvgPool}(\cdot)$ computes segment-wise means. This sub-task enables the agent to focus on slowly evolving components in the signal.

(2) Seasonality forecasting. Seasonal dynamics are extracted by removing the estimated trend from the original signal:

$$\mathbf{S} = \mathbf{Y} - \mathbf{F}, \quad \text{where} \quad \mathbf{F} = \mathbf{TrendModel}(\mathbf{Y}),$$

with $\mathbf{F}$ being a smoothed version of $\mathbf{Y}$, either computed or learned. This task guides the agent to attend to periodic or residual structures.

(3) Spectral energy forecasting. To encode frequency-domain characteristics, we apply the real-valued Fast Fourier Transform (FFT) and take the magnitude:

$$\mathbf{E} = |\mathbf{FFT}(\mathbf{Y})| \in \mathbb{R}^{(L//2+1)\times M}.$$

The resulting spectral energy profile provides insights into dominant frequencies and periodicities in the future signal.

(4) Statistical descriptor forecasting. We summarize $\mathbf{Y}$ using a fixed set of descriptive statistics, capturing distributional properties as follows:

$$\mathbf{D} = [\boldsymbol{\mu}, \boldsymbol{\sigma}, \boldsymbol{\gamma}, \boldsymbol{\kappa}, \mathbf{y}_{\max}, \mathbf{y}_{\min}] \in \mathbb{R}^{6\times M},$$

where

$$\boldsymbol{\mu} = \mathbf{Mean}(\mathbf{Y}), \quad \boldsymbol{\sigma} = \mathbf{Std}(\mathbf{Y}), \quad \boldsymbol{\gamma} = \mathbf{Skewness}(\mathbf{Y}), \quad \boldsymbol{\kappa} = \mathbf{Kurtosis}(\mathbf{Y}),$$

and

$$\mathbf{y}_{\max} = \max_t \mathbf{Y}[t], \quad \mathbf{y}_{\min} = \min_t \mathbf{Y}[t].$$

This task allows the agent to model global signal characteristics in a compact form.

## B  More Dataset Details

Table 3: Summary of the 11 datasets used in our experiments, covering diverse domains, temporal resolutions, and feature dimensions.

| Dataset | Variate | Input Length | Predict Length | Information | Frequency | Split |
|---|---|---|---|---|---|---|
| ETTh1 | 7 | $96 \sim 720$ | $96 \sim 720$ | Electricity | 15mins | 6:2:2 |
| ETTh2 | 7 | $96 \sim 720$ | $96 \sim 720$ | Electricity | 15mins | 6:2:2 |
| ETTm1 | 7 | $96 \sim 720$ | $96 \sim 720$ | Electricity | 15mins | 6:2:2 |
| ETTm2 | 7 | $96 \sim 720$ | $96 \sim 720$ | Electricity | 15mins | 6:2:2 |
| Weather | 21 | $96 \sim 720$ | $96 \sim 720$ | Environment | 10mins | 7:1:2 |
| Temperature | 108 | $96 \sim 720$ | $96 \sim 720$ | Environment | 3hours | 7:1:2 |
| AOShunyi | 11 | $96 \sim 720$ | $96 \sim 720$ | Environment | 1hour | 7:1:2 |
| AQWan | 11 | $96 \sim 720$ | $96 \sim 720$ | Environment | 1hour | 7:1:2 |
| PM2.5 | 108 | $96 \sim 720$ | $96 \sim 720$ | Nature | 3hours | 7:1:2 |
| ZafNoo | 11 | $96 \sim 720$ | $96 \sim 720$ | Nature | 30mins | 7:1:2 |
| CzeLan | 11 | $96 \sim 720$ | $96 \sim 720$ | Nature | 30mins | 7:1:2 |

As shown in Table 3, we evaluate our framework on 11 multivariate time series datasets spanning three real-world application domains: Electricity, Environment, and Nature. These datasets exhibit diverse characteristics in terms of feature dimensionality, sampling frequency, and domain semantics, offering a comprehensive benchmark for assessing forecasting performance under different temporal patterns and data complexities. The Electricity domain includes four ETT datasets: ETTh1, ETTh2, ETTm1, and ETTm2 [77, 60]. Each contains 7 variates recorded at 15-minute intervals, primarily reflecting industrial power consumption dynamics. The Environment domain covers Weather [60], Temperature [59], AQShunyi, and AQWan [41]. Weather contains 21 meteorological variables sampled every 10 minutes, while Temperature and PM2.5 have higher dimensionality (108 variates) at a coarser 3-hour frequency. AQShunyi and AQWan offer 11-dimensional hourly air quality readings from different regions in Beijing. The Nature domain consists of PM2.5 [59], ZafNoo, and

CzeLan [41], all of which represent long-range environmental trends collected over various regions, with sampling frequencies ranging from 30 minutes to 3 hours. Across all datasets, the input and prediction sequence lengths are consistently selected from the range 96 to 720 to ensure uniform temporal coverage across tasks. For the ETT datasets, we follow prior work and adopt a 6:2:2 split for training, validation, and testing. For the remaining datasets, we use a 7:1:2 split. This ensures both consistent evaluation and fair comparison with previous studies.

## C  More Implementation Details

All experiments are conducted on a server equipped with 8 NVIDIA A100 GPUs (80GB memory each). MAFS is implemented using PyTorch 1.13.0 [39] and optimized using the Adam optimizer [17] with an L2 loss. We investigate four distinct communication topologies for the agent interaction graph $\mathbf{G}$, including ring, star, chain, and fully-connected structures. Each forecasting agent is instantiated using the iTransformer architecture [28], with a unified configuration across all experiments: a fixed learning rate of 1e-3, a hidden dimension of 128, and 2 encoder layers. To evaluate the scalability of MAFS, we vary the number of agents $N \in \{4, 8, 12, 16, 20, 24\}$. MAFS training is conducted in two stages. In the first stage, each agent is independently optimized on its assigned sub-task using a learning rate of 1e-3 for 10 epochs. In the second stage, we freeze the parameters of all forecasting agents and jointly finetune only the topology-aware weight adjuster and the agent-rated voting aggregator. This stage also runs for 10 epochs, using a smaller learning rate of 1e-4 to enable stable convergence during topology adaptation and collaborative forecasting. For all tasks, we adopt a symmetric prediction setting where the historical input length equals the forecasting horizon, following the protocol in [76], to enable consistent evaluation across datasets. To ensure fair comparison, we re-run iTransformer under our unified setting (including learning rate, hidden size, and number of layers). For all baselines other than iTransformer, we use reported results from the iTransformer and TFB [41] papers, except for the `temp` and `pm2.5` datasets, where we re-run all baseline models. Importantly, we identified and corrected a bug in the original evaluation process: the `test_loader` was configured with `drop_last=True` during testing, which led to the exclusion of final test batches. We explicitly set `drop_last=False` to ensure fair and complete evaluation [41].

## D  Full Results of Comparison Between Two Types of Sub-tasks Division

Figure 8 to Figure 18 present a comprehensive comparison of homogeneous-correlated and heterogeneous-correlated sub-task configurations across all 11 datasets. The results demonstrate that each configuration exhibits distinct advantages under different data characteristics, emphasizing the need for adaptive task decomposition in multi-agent forecasting.

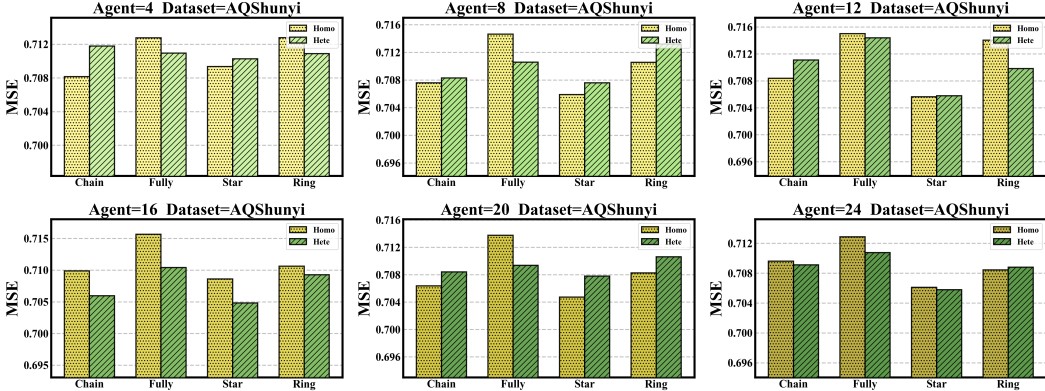

Figure 8: Performance comparison of homogeneous-correlated and heterogeneous-correlated sub-tasks on AQShunyi.

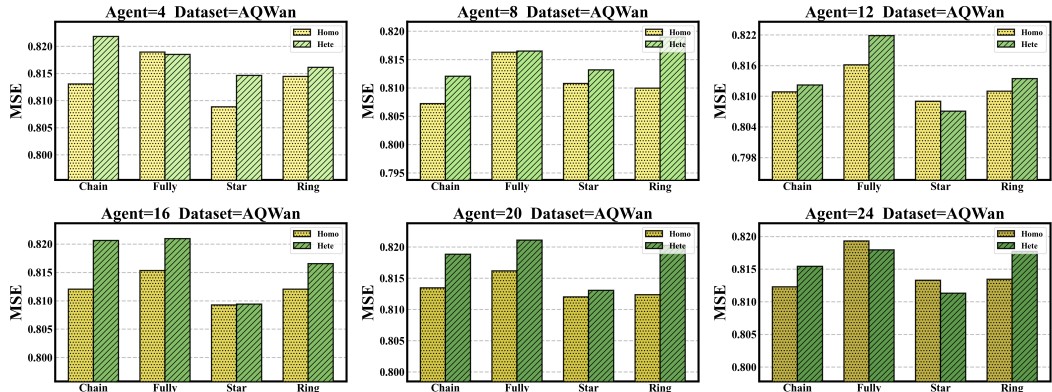

Figure 9: Performance comparison of homogeneous-correlated and heterogeneous-correlated sub-tasks on AQWan.

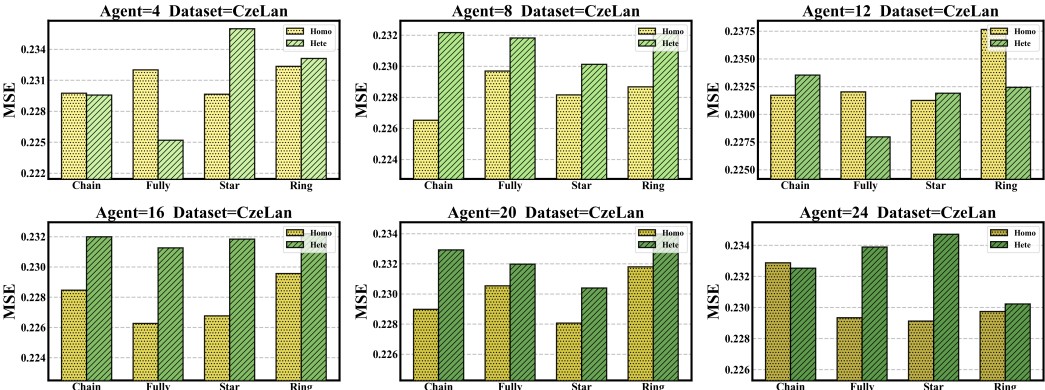

Figure 10: Performance comparison of homogeneous-correlated and heterogeneous-correlated sub-tasks on CzeLan.

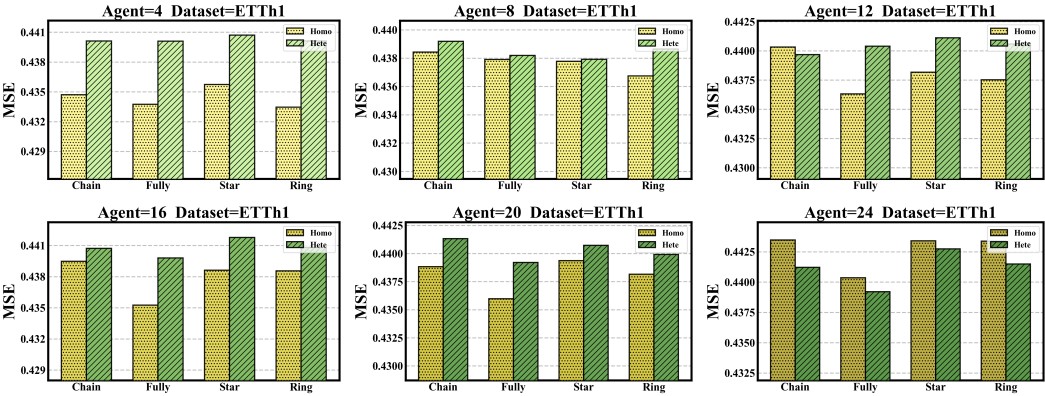

Figure 11: Performance comparison of homogeneous-correlated and heterogeneous-correlated sub-tasks on ETTh1.

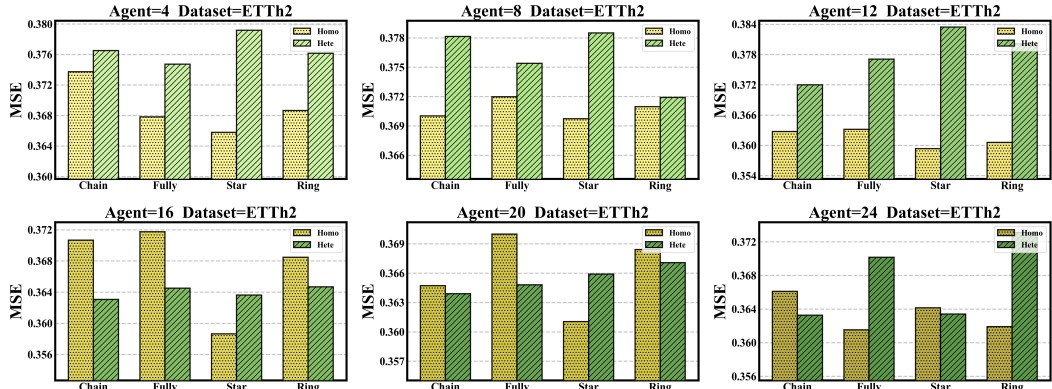

Figure 12: Performance comparison of homogeneous-correlated and heterogeneous-correlated sub-tasks on ETTh2.

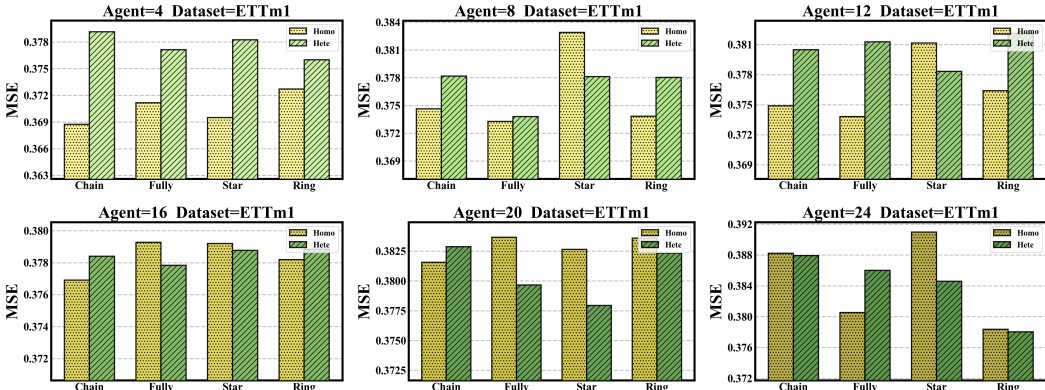

Figure 13: Performance comparison of homogeneous-correlated and heterogeneous-correlated sub-tasks on ETTm1.

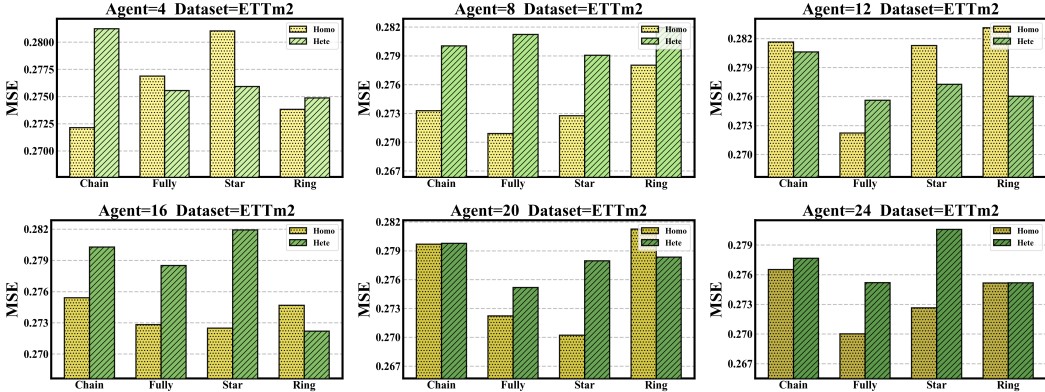

Figure 14: Performance comparison of homogeneous-correlated and heterogeneous-correlated sub-tasks on ETTm2.

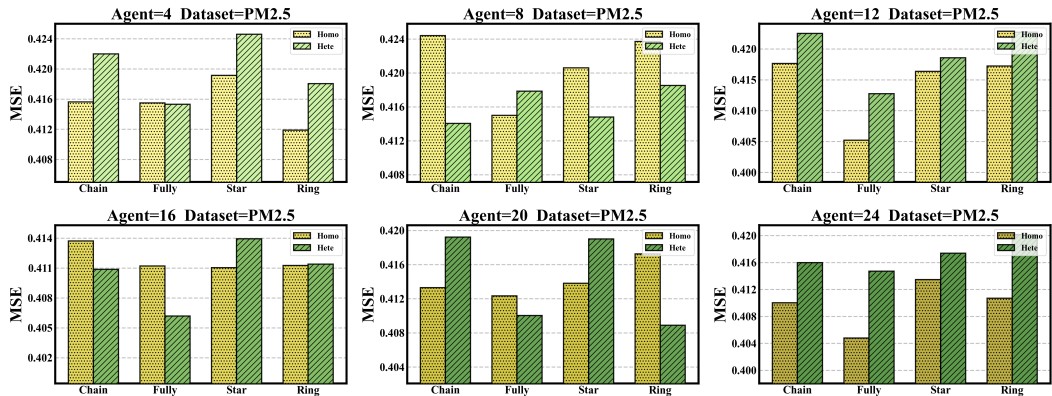

Figure 15: Performance comparison of homogeneous-correlated and heterogeneous-correlated sub-tasks on pm2.5.

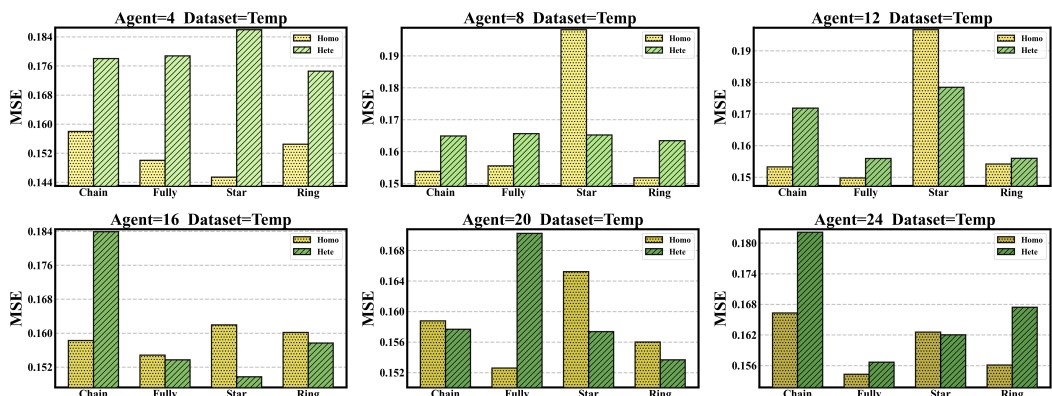

Figure 16: Performance comparison of homogeneous-correlated and heterogeneous-correlated sub-tasks on temp.

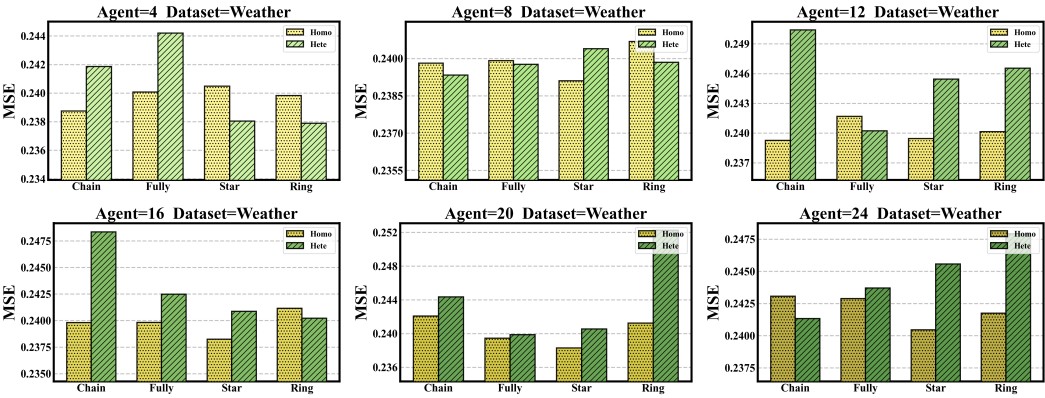

Figure 17: Performance comparison of homogeneous-correlated and heterogeneous-correlated sub-tasks on weather.

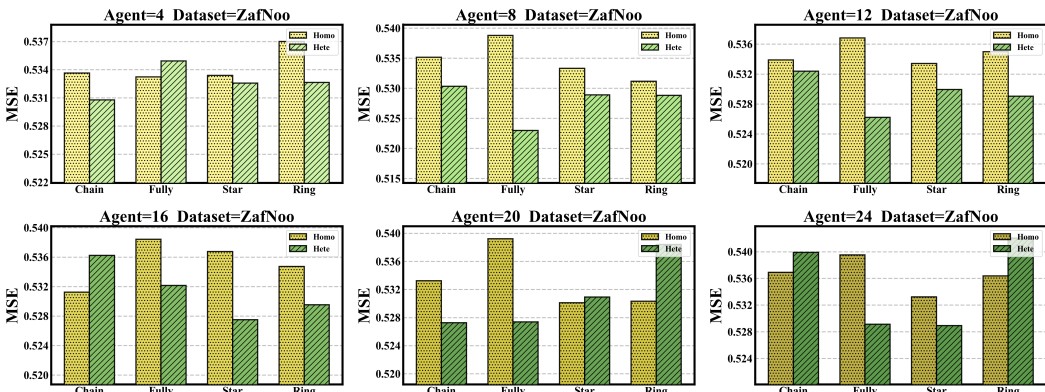

Figure 18: Performance comparison of homogeneous-correlated and heterogeneous-correlated sub-tasks on ZafNoo.

# E  Full Results of Different MAFS Scales

Figure 19 to Figure 29 show the full results of MAFS under varying numbers of agents across all datasets. Overall, we observe that increasing the number of agents generally leads to more stable forecasting performance, with reduced variance across different runs. Although more agents do not always yield the best accuracy, the ensemble effect tends to enhance robustness. This finding supports the scalability of MAFS in diverse time series scenarios and highlights the trade-off between performance and computational overhead when scaling agent populations.

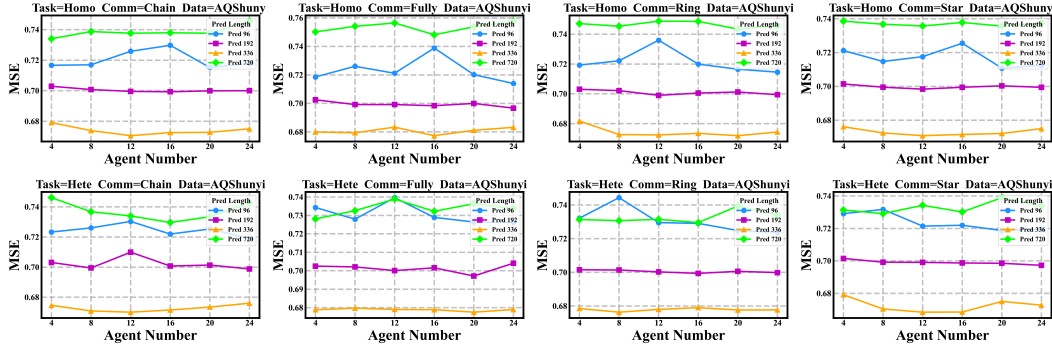

Figure 19: Performance comparison of different agent numbers on AQShunyi.

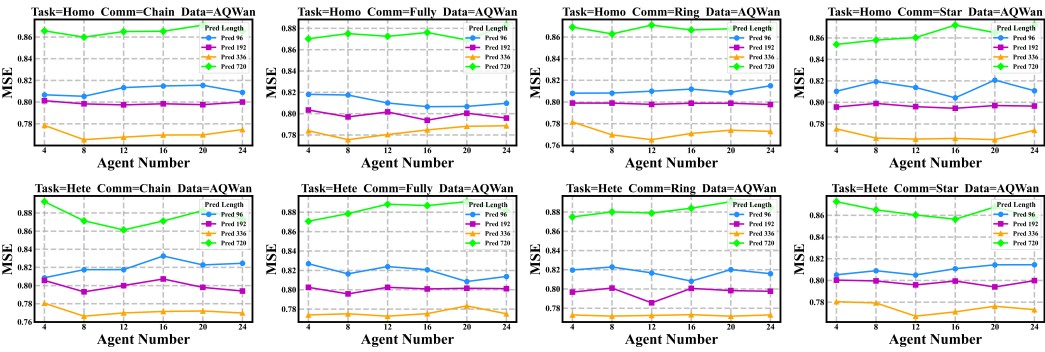

Figure 20: Performance comparison of different agent numbers on AQWan.

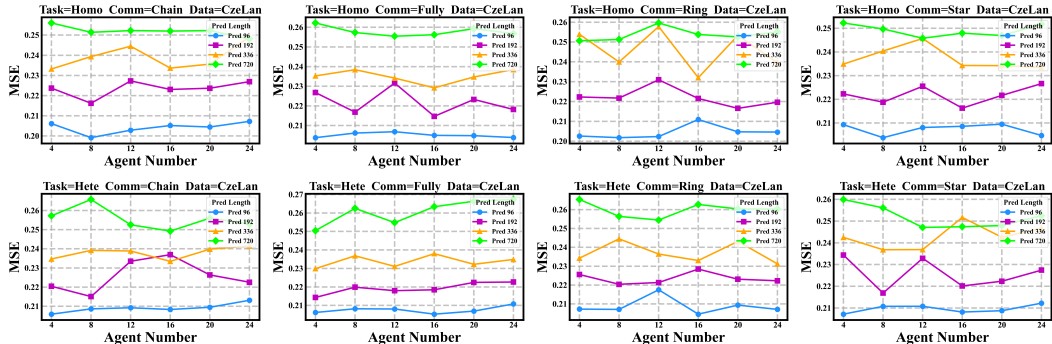

Figure 21: Performance comparison of different agent numbers on CzeLan.

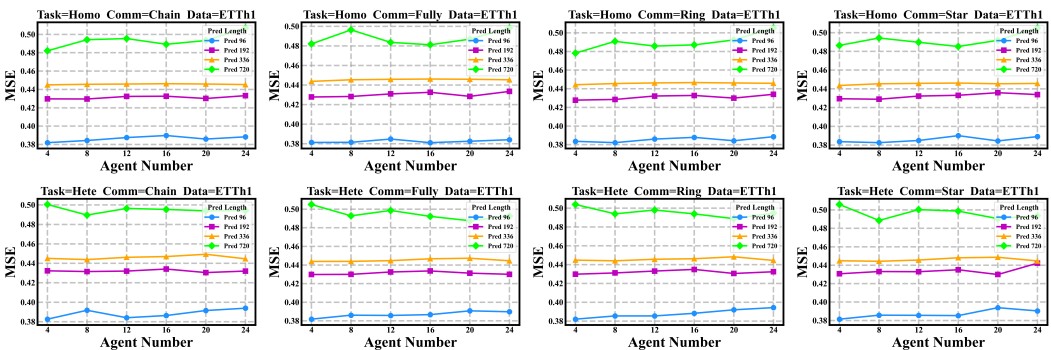

Figure 22: Performance comparison of different agent numbers on ETTh1.

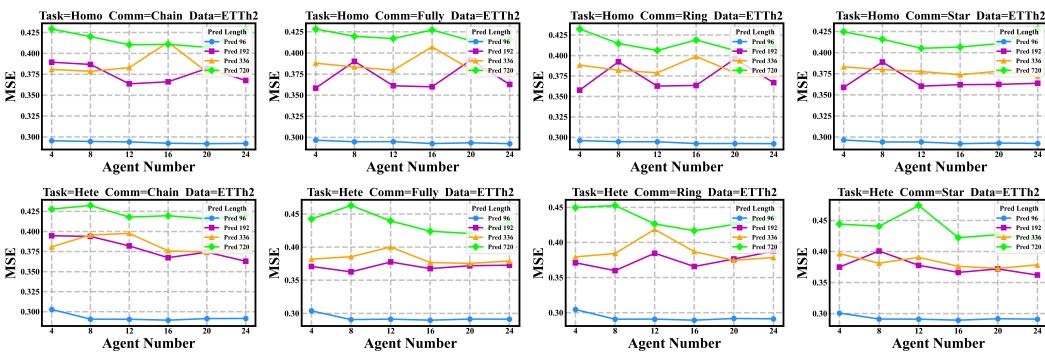

Figure 23: Performance comparison of different agent numbers on ETTh2.

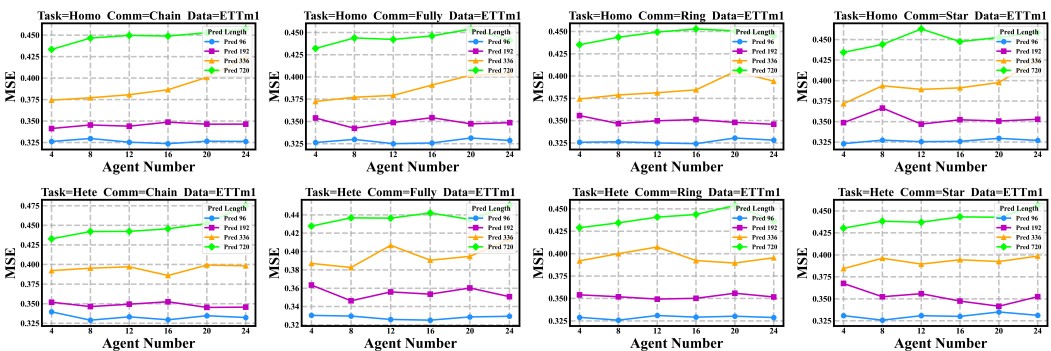

Figure 24: Performance comparison of different agent numbers on ETTm1.

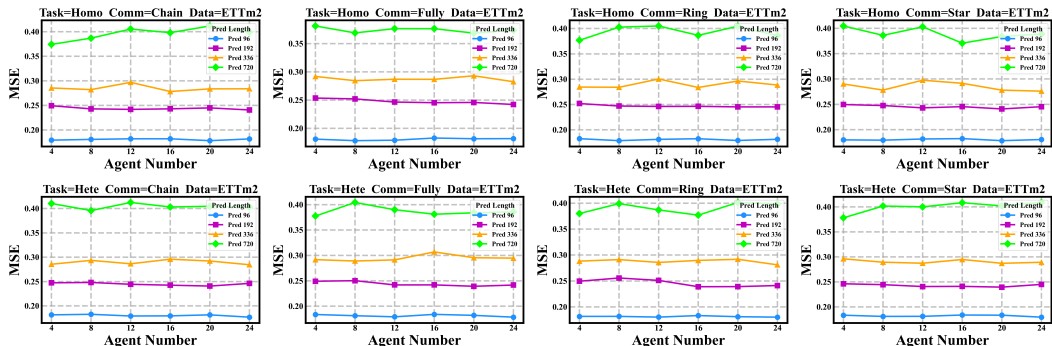

Figure 25: Performance comparison of different agent numbers on ETTm2.

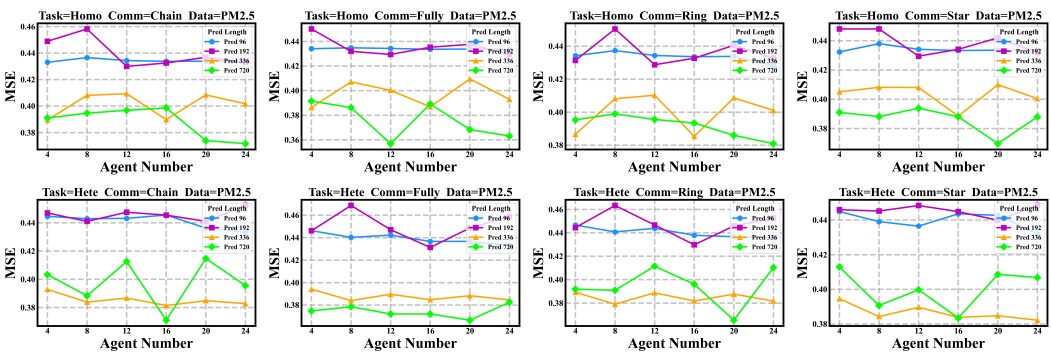

Figure 26: Performance comparison of different agent numbers on pm2.5.

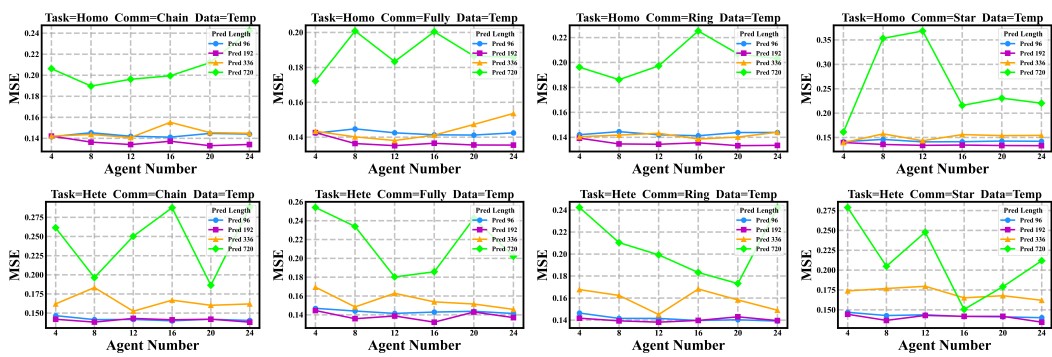

Figure 27: Performance comparison of different agent numbers on temp.

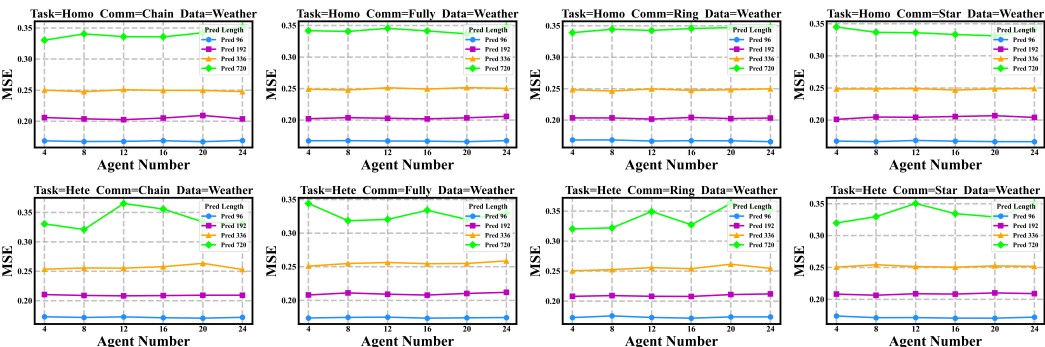

Figure 28: Performance comparison of different agent numbers on weather.

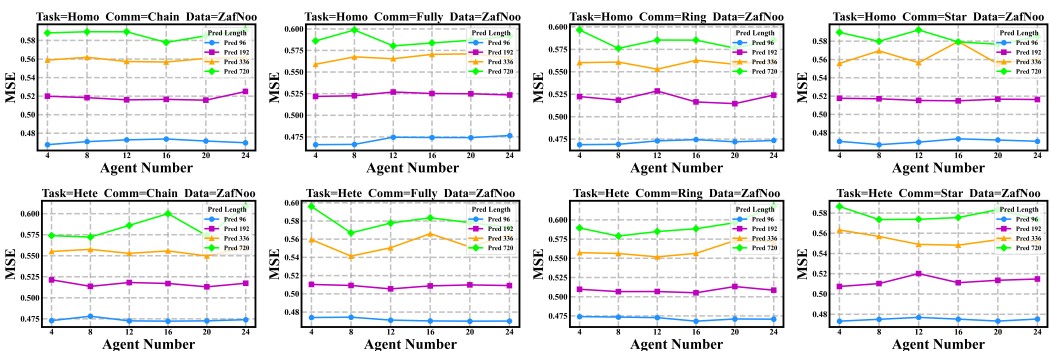

Figure 29: Performance comparison of different agent numbers on ZafNoo.

# F    Limitation

This work introduces MAFS, a novel multi-agent forecasting system that successfully leverages collective intelligence to enhance long-term time series forecasting. Through specialized agent design, structured communication, and adaptive decision fusion, MAFS achieves consistent performance improvements across diverse datasets and forecasting horizons, demonstrating its strong generalization and robustness. However, despite these promising results, multi-agent forecasting still faces certain limitations. Specifically, the system exhibits occasional instability during training, particularly when the number of agents increases or when dealing with highly volatile datasets. This instability may arise from conflicting agent objectives or suboptimal communication structures that hinder effective information integration.

In future work, we plan to further improve MAFS from the following perspectives:

- **Adaptive Communication Topologies**: Develop dynamic topology learning mechanisms that can automatically adjust inter-agent connections based on data characteristics and task complexity, reducing reliance on manually defined structures.
- **Diverse Task Decomposition Strategies**: Explore more fine-grained and semantically meaningful task divisions, enabling agents to specialize in richer forecasting aspects such as uncertainty estimation, anomaly detection, or rare event prediction.

We believe these directions will further enhance the stability, adaptability, and forecasting capability of multi-agent systems in real-world scenarios.

# G  Full Long-term Time Series Forecasting Results

Table 4: Comparison of long-term time series forecasting methods on 11 datasets using MSE and MAE (lower is better). Best results are marked in red ; second-best results are underlined in blue .

| Methods | | MAFS (Ours) | | iTransformer [2024] | | TimeMixer [2024] | | PatchTST [2024] | | Crossformer [2023] | | TiDE [2024] | | TimesNet [2023] | | DLinear [2023] | | Autoformer [2021] | | Informer [2021] | |
|---|---|---|---|---|---|---|---|---|---|---|---|---|---|---|---|---|---|---|---|---|---|
| Datasets | Metric | MSE | MAE | MSE | MAE | MSE | MAE | MSE | MAE | MSE | MAE | MSE | MAE | MSE | MAE | MSE | MAE | MSE | MAE | MSE | MAE |
| ETTh1 | 96 | 0.381 | 0.399 | 0.392 | 0.41 | 0.375 | 0.4 | 0.414 | 0.419 | 0.423 | 0.448 | 0.479 | 0.464 | 0.384 | 0.402 | 0.386 | 0.4 | 0.449 | 0.459 | 0.865 | 0.713 |
| | 192 | 0.428 | 0.424 | 0.442 | 0.442 | 0.429 | 0.421 | 0.46 | 0.445 | 0.471 | 0.474 | 0.525 | 0.492 | 0.436 | 0.429 | 0.437 | 0.432 | 0.5 | 0.482 | 1.008 | 0.792 |
| | 336 | 0.444 | 0.442 | 0.483 | 0.473 | 0.484 | 0.458 | 0.501 | 0.466 | 0.57 | 0.546 | 0.565 | 0.515 | 0.491 | 0.469 | 0.481 | 0.459 | 0.521 | 0.496 | 1.107 | 0.809 |
| | 720 | 0.478 | 0.484 | 0.552 | 0.537 | 0.498 | 0.482 | 0.5 | 0.488 | 0.653 | 0.621 | 0.594 | 0.558 | 0.521 | 0.5 | 0.519 | 0.516 | 0.514 | 0.512 | 1.181 | 0.865 |
| | Avg | 0.433 | 0.437 | 0.467 | 0.466 | 0.447 | 0.44 | 0.469 | 0.454 | 0.529 | 0.522 | 0.541 | 0.507 | 0.458 | 0.45 | 0.456 | 0.452 | 0.496 | 0.487 | 1.04 | 0.795 |
| ETTh2 | 96 | 0.289 | 0.341 | 0.304 | 0.353 | 0.289 | 0.341 | 0.302 | 0.348 | 0.745 | 0.584 | 0.4 | 0.44 | 0.34 | 0.374 | 0.333 | 0.387 | 0.346 | 0.388 | 3.755 | 1.525 |
| | 192 | 0.358 | 0.391 | 0.399 | 0.417 | 0.372 | 0.392 | 0.388 | 0.4 | 0.877 | 0.656 | 0.528 | 0.509 | 0.402 | 0.414 | 0.477 | 0.476 | 0.456 | 0.452 | 5.602 | 1.931 |
| | 336 | 0.372 | 0.406 | 0.41 | 0.429 | 0.386 | 0.414 | 0.426 | 0.433 | 1.043 | 0.731 | 0.643 | 0.571 | 0.452 | 0.452 | 0.594 | 0.541 | 0.482 | 0.486 | 4.721 | 1.835 |
| | 720 | 0.405 | 0.438 | 0.433 | 0.463 | 0.412 | 0.434 | 0.431 | 0.446 | 1.104 | 0.763 | 0.874 | 0.679 | 0.462 | 0.468 | 0.831 | 0.657 | 0.515 | 0.511 | 3.647 | 1.625 |
| | Avg | 0.356 | 0.394 | 0.386 | 0.415 | 0.364 | 0.395 | 0.387 | 0.407 | 0.942 | 0.684 | 0.611 | 0.55 | 0.414 | 0.427 | 0.559 | 0.515 | 0.45 | 0.459 | 4.431 | 1.729 |
| ETTm1 | 96 | 0.323 | 0.361 | 0.336 | 0.372 | 0.32 | 0.357 | 0.329 | 0.367 | 0.404 | 0.426 | 0.364 | 0.387 | 0.338 | 0.375 | 0.345 | 0.372 | 0.505 | 0.475 | 0.672 | 0.571 |
| | 192 | 0.341 | 0.373 | 0.366 | 0.39 | 0.361 | 0.381 | 0.367 | 0.385 | 0.45 | 0.451 | 0.398 | 0.404 | 0.374 | 0.387 | 0.38 | 0.389 | 0.553 | 0.496 | 0.795 | 0.669 |
| | 336 | 0.372 | 0.393 | 0.388 | 0.41 | 0.39 | 0.404 | 0.399 | 0.41 | 0.532 | 0.515 | 0.428 | 0.425 | 0.41 | 0.411 | 0.413 | 0.413 | 0.621 | 0.537 | 1.212 | 0.871 |
| | 720 | 0.428 | 0.426 | 0.442 | 0.442 | 0.454 | 0.441 | 0.454 | 0.439 | 0.666 | 0.589 | 0.487 | 0.461 | 0.478 | 0.45 | 0.474 | 0.453 | 0.671 | 0.561 | 1.166 | 0.823 |
| | Avg | 0.366 | 0.388 | 0.383 | 0.403 | 0.381 | 0.395 | 0.387 | 0.4 | 0.513 | 0.496 | 0.419 | 0.419 | 0.4 | 0.406 | 0.403 | 0.407 | 0.588 | 0.517 | 0.961 | 0.734 |
| ETTm2 | 96 | 0.177 | 0.26 | 0.184 | 0.266 | 0.175 | 0.258 | 0.175 | 0.259 | 0.287 | 0.366 | 0.207 | 0.305 | 0.187 | 0.267 | 0.193 | 0.292 | 0.255 | 0.339 | 0.365 | 0.453 |
| | 192 | 0.239 | 0.302 | 0.256 | 0.317 | 0.237 | 0.299 | 0.241 | 0.302 | 0.414 | 0.492 | 0.29 | 0.364 | 0.249 | 0.309 | 0.284 | 0.362 | 0.281 | 0.34 | 0.533 | 0.563 |
| | 336 | 0.276 | 0.329 | 0.313 | 0.355 | 0.298 | 0.34 | 0.305 | 0.343 | 0.597 | 0.542 | 0.377 | 0.422 | 0.321 | 0.351 | 0.369 | 0.427 | 0.339 | 0.372 | 1.363 | 0.887 |
| | 720 | 0.369 | 0.395 | 0.407 | 0.417 | 0.391 | 0.396 | 0.402 | 0.4 | 1.73 | 1.042 | 0.558 | 0.524 | 0.408 | 0.403 | 0.554 | 0.522 | 0.433 | 0.432 | 3.379 | 1.338 |
| | Avg | 0.265 | 0.321 | 0.29 | 0.339 | 0.275 | 0.323 | 0.281 | 0.326 | 0.757 | 0.61 | 0.358 | 0.404 | 0.291 | 0.333 | 0.35 | 0.401 | 0.327 | 0.371 | 1.41 | 0.81 |
| Weather | 96 | 0.166 | 0.206 | 0.175 | 0.215 | 0.163 | 0.209 | 0.177 | 0.218 | 0.158 | 0.23 | 0.202 | 0.261 | 0.172 | 0.22 | 0.196 | 0.255 | 0.266 | 0.336 | 0.3 | 0.384 |
| | 192 | 0.201 | 0.244 | 0.214 | 0.254 | 0.208 | 0.25 | 0.225 | 0.259 | 0.206 | 0.277 | 0.242 | 0.298 | 0.219 | 0.261 | 0.237 | 0.296 | 0.307 | 0.367 | 0.598 | 0.544 |
| | 336 | 0.246 | 0.282 | 0.252 | 0.288 | 0.251 | 0.287 | 0.278 | 0.297 | 0.272 | 0.335 | 0.287 | 0.335 | 0.28 | 0.306 | 0.283 | 0.335 | 0.359 | 0.395 | 0.578 | 0.523 |
| | 720 | 0.318 | 0.338 | 0.331 | 0.348 | 0.339 | 0.341 | 0.354 | 0.348 | 0.398 | 0.418 | 0.351 | 0.386 | 0.365 | 0.359 | 0.345 | 0.381 | 0.419 | 0.428 | 1.059 | 0.741 |
| | Avg | 0.233 | 0.267 | 0.243 | 0.276 | 0.24 | 0.271 | 0.259 | 0.281 | 0.259 | 0.315 | 0.271 | 0.32 | 0.259 | 0.287 | 0.265 | 0.317 | 0.338 | 0.382 | 0.634 | 0.548 |
| AQShunyi | 96 | 0.711 | 0.499 | 0.742 | 0.506 | 0.731 | 0.533 | 0.648 | 0.481 | 0.652 | 0.484 | 0.708 | 0.52 | 0.658 | 0.488 | 0.651 | 0.492 | 0.736 | 0.529 | 0.754 | 0.542 |
| | 192 | 0.697 | 0.502 | 0.71 | 0.507 | 0.711 | 0.467 | 0.69 | 0.501 | 0.674 | 0.499 | 0.774 | 0.569 | 0.707 | 0.511 | 0.691 | 0.512 | 0.735 | 0.535 | 0.759 | 0.536 |
| | 336 | 0.668 | 0.506 | 0.687 | 0.51 | 0.684 | 0.564 | 0.711 | 0.515 | 0.704 | 0.515 | 0.827 | 0.56 | 0.785 | 0.537 | 0.716 | 0.529 | 0.83 | 0.566 | 0.837 | 0.56 |
| | 720 | 0.728 | 0.533 | 0.752 | 0.539 | 0.749 | 0.554 | 0.77 | 0.538 | 0.747 | 0.518 | 0.803 | 0.566 | 0.755 | 0.527 | 0.765 | 0.556 | 0.754 | 0.532 | 0.777 | 0.543 |
| | Avg | 0.701 | 0.509 | 0.723 | 0.515 | 0.719 | 0.529 | 0.705 | 0.509 | 0.694 | 0.504 | 0.778 | 0.554 | 0.726 | 0.516 | 0.706 | 0.522 | 0.764 | 0.541 | 0.782 | 0.545 |
| AQWan | 96 | 0.804 | 0.49 | 0.814 | 0.491 | 0.829 | 0.456 | 0.745 | 0.47 | 0.75 | 0.465 | 0.833 | 0.524 | 0.791 | 0.488 | 0.756 | 0.481 | 0.858 | 0.518 | 0.901 | 0.522 |
| | 192 | 0.786 | 0.496 | 0.801 | 0.497 | 0.81 | 0.501 | 0.792 | 0.491 | 0.762 | 0.479 | 0.82 | 0.516 | 0.779 | 0.49 | 0.8 | 0.502 | 0.803 | 0.513 | 0.833 | 0.521 |
| | 336 | 0.765 | 0.496 | 0.786 | 0.502 | 0.791 | 0.538 | 0.819 | 0.503 | 0.802 | 0.504 | 0.858 | 0.552 | 0.814 | 0.505 | 0.823 | 0.516 | 0.826 | 0.523 | 0.847 | 0.525 |
| | 720 | 0.854 | 0.529 | 0.868 | 0.536 | 0.883 | 0.499 | 0.89 | 0.533 | 0.83 | 0.511 | 0.913 | 0.551 | 0.869 | 0.519 | 0.891 | 0.548 | 0.872 | 0.547 | 0.883 | 0.532 |
| | Avg | 0.802 | 0.503 | 0.817 | 0.507 | 0.828 | 0.499 | 0.812 | 0.499 | 0.786 | 0.49 | 0.856 | 0.536 | 0.813 | 0.5 | 0.818 | 0.512 | 0.84 | 0.525 | 0.866 | 0.525 |
| CzeLan | 96 | 0.199 | 0.248 | 0.21 | 0.255 | 0.202 | 0.263 | 0.183 | 0.251 | 0.581 | 0.443 | 0.186 | 0.256 | 0.176 | 0.237 | 0.211 | 0.289 | 0.238 | 0.294 | 0.25 | 0.305 |
| | 192 | 0.214 | 0.258 | 0.231 | 0.275 | 0.22 | 0.288 | 0.208 | 0.271 | 0.705 | 0.503 | 0.226 | 0.29 | 0.215 | 0.279 | 0.252 | 0.323 | 0.29 | 0.341 | 0.295 | 0.337 |
| | 336 | 0.229 | 0.281 | 0.243 | 0.293 | 0.237 | 0.266 | 0.243 | 0.302 | 0.971 | 0.596 | 0.238 | 0.304 | 0.224 | 0.288 | 0.317 | 0.366 | 0.322 | 0.357 | 0.335 | 0.361 |
| | 720 | 0.246 | 0.298 | 0.245 | 0.297 | 0.254 | 0.302 | 0.273 | 0.335 | 1.566 | 0.762 | 0.295 | 0.363 | 0.282 | 0.337 | 0.358 | 0.392 | 0.379 | 0.427 | 0.384 | 0.416 |
| | Avg | 0.222 | 0.271 | 0.232 | 0.28 | 0.228 | 0.28 | 0.227 | 0.29 | 0.956 | 0.576 | 0.237 | 0.303 | 0.224 | 0.285 | 0.284 | 0.342 | 0.307 | 0.355 | 0.316 | 0.355 |
| ZafNoo | 96 | 0.466 | 0.411 | 0.476 | 0.419 | 0.481 | 0.404 | 0.444 | 0.426 | 0.432 | 0.419 | 0.508 | 0.45 | 0.479 | 0.424 | 0.434 | 0.411 | 0.524 | 0.468 | 0.541 | 0.473 |
| | 192 | 0.505 | 0.439 | 0.529 | 0.457 | 0.527 | 0.454 | 0.498 | 0.456 | 0.479 | 0.449 | 0.536 | 0.491 | 0.491 | 0.446 | 0.484 | 0.444 | 0.687 | 0.558 | 0.708 | 0.575 |
| | 336 | 0.541 | 0.465 | 0.568 | 0.488 | 0.56 | 0.444 | 0.53 | 0.48 | 0.521 | 0.469 | 0.592 | 0.519 | 0.551 | 0.479 | 0.518 | 0.464 | 0.835 | 0.669 | 0.851 | 0.661 |
| | 720 | 0.567 | 0.494 | 0.591 | 0.509 | 0.585 | 0.46 | 0.574 | 0.499 | 0.543 | 0.483 | 0.642 | 0.533 | 0.627 | 0.511 | 0.548 | 0.486 | 0.854 | 0.702 | 0.876 | 0.699 |
| | Avg | 0.52 | 0.451 | 0.541 | 0.468 | 0.538 | 0.44 | 0.511 | 0.465 | 0.494 | 0.455 | 0.569 | 0.498 | 0.537 | 0.465 | 0.496 | 0.451 | 0.725 | 0.599 | 0.744 | 0.602 |
| PM2.5 | 96 | 0.428 | 0.428 | 0.438 | 0.432 | 0.446 | 0.48 | 0.46 | 0.468 | 0.451 | 0.46 | 0.475 | 0.484 | 0.481 | 0.481 | 0.453 | 0.466 | 0.525 | 0.512 | 0.581 | 0.594 |
| | 192 | 0.429 | 0.425 | 0.436 | 0.429 | 0.45 | 0.408 | 0.462 | 0.449 | 0.463 | 0.446 | 0.493 | 0.476 | 0.455 | 0.446 | 0.447 | 0.431 | 0.519 | 0.502 | 0.55 | 0.537 |
| | 336 | 0.379 | 0.404 | 0.412 | 0.417 | 0.395 | 0.441 | 0.466 | 0.505 | 0.447 | 0.469 | 0.455 | 0.473 | 0.469 | 0.508 | 0.451 | 0.485 | 0.53 | 0.553 | 0.524 | 0.546 |
| | 720 | 0.357 | 0.399 | 0.4 | 0.408 | 0.37 | 0.414 | 0.452 | 0.493 | 0.464 | 0.514 | 0.499 | 0.554 | 0.49 | 0.535 | 0.458 | 0.524 | 0.485 | 0.53 | 0.499 | 0.566 |
| | Avg | 0.398 | 0.414 | 0.421 | 0.421 | 0.415 | 0.436 | 0.46 | 0.479 | 0.456 | 0.472 | 0.481 | 0.497 | 0.473 | 0.492 | 0.453 | 0.477 | 0.515 | 0.524 | 0.539 | 0.561 |
| Temp | 96 | 0.139 | 0.282 | 0.138 | 0.283 | 0.144 | 0.306 | 0.144 | 0.3 | 0.186 | 0.377 | 0.155 | 0.323 | 0.162 | 0.329 | 0.152 | 0.317 | 0.185 | 0.373 | 0.197 | 0.405 |
| | 192 | 0.132 | 0.279 | 0.145 | 0.296 | 0.137 | 0.317 | 0.145 | 0.309 | 0.203 | 0.435 | 0.157 | 0.323 | 0.173 | 0.374 | 0.155 | 0.322 | 0.213 | 0.45 | 0.199 | 0.429 |
| | 336 | 0.138 | 0.289 | 0.203 | 0.352 | 0.145 | 0.297 | 0.147 | 0.313 | 0.209 | 0.435 | 0.169 | 0.357 | 0.185 | 0.387 | 0.162 | 0.342 | 0.231 | 0.492 | 0.243 | 0.506 |
| | 720 | 0.151 | 0.3 | 0.204 | 0.352 | 0.157 | 0.298 | 0.153 | 0.303 | 0.224 | 0.445 | 0.173 | 0.35 | 0.312 | 0.623 | 0.167 | 0.329 | 0.347 | 0.683 | 0.311 | 0.627 |
| | Avg | 0.14 | 0.288 | 0.173 | 0.321 | 0.146 | 0.304 | 0.147 | 0.306 | 0.206 | 0.423 | 0.164 | 0.338 | 0.208 | 0.428 | 0.159 | 0.327 | 0.244 | 0.5 | 0.238 | 0.492 |

