# OpenReview forum: "Many Minds, One Goal: Time Series Forecasting via Sub-task Specialization and Inter-agent Cooperation"
_NeurIPS.cc/2025/Conference — NeurIPS 2025 poster_

### Official Review · Reviewer_PB78 · 2025-07-01

**Clarity:** 3
**Significance:** 3
**Originality:** 4
**Rating:** 5
**Confidence:** 4

**Summary:**

This paper proposes a novel and effective framework, the Multi-Agent Forecasting System (MAFS), to address the inherent diversity and complexity of time series forecasting tasks. Unlike traditional single-model approaches, MAFS decomposes the problem into multiple specialized agents, each focusing on different temporal patterns or signal characteristics. These agents collaborate through flexible communication topologies and their outputs are aggregated via a lightweight voting mechanism. The approach is well-motivated, intuitive, and supported by extensive experiments across 11 benchmark datasets, demonstrating its superior robustness and adaptability compared to conventional forecasting models.

**Questions:**

Q1. Why was iTransformer chosen as the backbone for the agents in MAFS? Have other backbone networks been tried?

Q2. The communication structures between agents are limited to graph/star/ring/chain. What is the rationale behind this selection? Are there other possible communication topologies worth exploring?

Q3. For the distribution-focused agent, why was the design to predict six separate statistical values instead of directly modeling the future data distribution?

**Ethical Concerns:**

["NO or VERY MINOR ethics concerns only"]

**Limitations:**

Yes, the authors have properly discussed the potential limitations of this research.

**Quality:**

3

**Strengths And Weaknesses:**

**Strengths:**

S1. Exploring a multi-agent system for time-series forecasting tasks is a novel and meaningful attempt, as forecasting inherently involves multiple factors and randomness. The proposed MAFS framework with multi-role agents improves the robustness of prediction process.

S2. MAFS decomposes the forecasting task into trend, seasonality, frequency, and distribution agents, which intuitively makes sense and enhances interpretability.

S3. The experimental results are comprehensive, which validate the consistent effectiveness of MAFS across 11 real-world datasets, and further verifies the robustness and adaptiveness of the multi-agent system across various datasets. The ablation studies confirm that the multi-agent system outperforms single-model baselines.

S4. The paper is well-structured, with clear figures and strong readability.

**Weaknesses:**

W1. MAFS performs worse than Crossformer on the AQShunyi and AQWan datasets, and the authors have not provided a detailed analysis of the reasons behind this.

W2. There are two typos in the paper: line 270 "MAS forecasting system" and line 194 "H^{L} \_i", which should be corrected. In addition, the decimal places in Table 1 should be consistently kept to three digits.

W3. In Section 5.7 (Case Study), the learned interaction weights among agents are visualized, but it appears that self-interaction weights are consistently dominant. The paper lacks an analysis of this phenomenon.

---

> ### Author Rebuttal · Authors · 2025-07-31
>
> Dear Reviewer PB78,
>
> Thank you very much for your valuable comments and positive support for our work. We will address your questions and concerns point by point.
>
> ---
> >**W1. More Detailed Analysis of Results**
>
> Thank you for the insightful comment. The relatively better performance of Crossformer on AQShunyi and AQWan may be attributed to its strength in jointly modeling temporal and inter-variable dependencies through attention mechanisms. In contrast, MAFS is instantiated with iTransformer in this experiment, which may have a comparatively weaker capacity to capture complex variable-wise correlations. Moreover, these two datasets exhibit strong global correlation patterns, where Crossformer's holistic modeling approach is particularly effective.
>
> **However, we would like to emphasize that MAFS is a model-agnostic framework. As demonstrated in our response to Q1, the agents in MAFS can be instantiated using different backbone models, and still lead to improved forecasting performance.** We will include this clarification in our revised version to better explain the observed performance gap.
>
> ---
> >**W2. Minor Typos**
>
> Thank you for pointing this out. We have corrected the typos at line 270 (“MAS” → “MAFS”) and line 194 (“H^{L} _i” → properly formatted). We have also ensured that all decimal values in Table 1 are consistently rounded to three digits. These changes will be reflected in the revised version.
>
> ---
> >**W3. More analysis of Figure 5**
>
> Thank you for the insightful comment. We observe that the dominance of self-interaction weights reflects each agent's specialization in distinct forecasting aspects. **Since each agent is assigned a decomposed task (e.g., specific variable groups or temporal patterns), it tends to rely more on its own features when making predictions. However, cross-agent interactions still play a supportive role, especially in capturing shared patterns across tasks.** We will add this analysis to the revised version to clarify the observed attention distribution.
>
> ---
> >**Q1. Why was iTransformer chosen as the backbone for the agents in MAFS? Have other backbone networks been tried?**
>
> We chose **iTransformer** as the default backbone for MAFS because it represents a clean and standard Transformer-based architecture, without specialized designs. This makes it a suitable foundation to clearly demonstrate the benefits introduced by the MAFS multi-agent structure, without interference from model-specific enhancements.
>
> To verify the generality of MAFS, we also integrated it with other representative backbones from different model families, including:
>
> * **PatchTST** (former-based)
> * **TimeMixer** (linear-based)
> * **SegRNN** (RNN-based)
>
> Across all 11 datasets, MAFS consistently outperforms their single-model counterparts, as shown below:
>
>
>
> |  Dataset |  MAFS(PatchTST) | Single(PatchTST) |
> | :------: | :-------------: | :--------------: |
> |   ETTh1  | **0.437 0.439** |    0.469 0.455   |
> |   ETTh2  | **0.373 0.394** |    0.387 0.407   |
> |   ETTm1  | **0.371 0.390** |    0.387 0.400   |
> |   ETTm2  | **0.265 0.318** |    0.281 0.326   |
> |  weather | **0.231 0.272** |    0.259 0.280   |
> | AQShunyi | **0.678 0.496** |    0.705 0.509   |
> |   AQWan  | **0.774 0.483** |    0.812 0.499   |
> |  CzeLan  | **0.217 0.273** |    0.227 0.290   |
> |  ZafNoo  | **0.494 0.453** |    0.512 0.465   |
> |  pm2\_5  | **0.407 0.420** |    0.460 0.479   |
> |   temp   | **0.142 0.289** |    0.147 0.306   |
>
>
>
> |  Dataset | MAFS(TimeMixer) | Single(TimeMixer) |
> | :------: | :-------------: | :---------------: |
> |   ETTh1  | **0.434 0.426** |     0.447 0.44    |
> |   ETTh2  | **0.353 0.382** |    0.365 0.395    |
> |   ETTm1  | **0.368 0.381** |    0.381 0.396    |
> |   ETTm2  | **0.264 0.314** |    0.275 0.323    |
> |  weather | **0.231 0.264** |     0.24 0.272    |
> | AQShunyi | **0.695 0.502** |     0.719 0.53    |
> |   AQWan  | **0.781 0.478** |    0.828 0.499    |
> |  CzeLan  |  **0.22 0.273** |     0.228 0.28    |
> |  ZafNoo  | **0.518 0.425** |    0.538 0.441    |
> |  pm2\_5  | **0.401 0.416** |    0.415 0.436    |
> |   temp   | **0.139 0.291** |    0.146 0.305    |
>
>
>
> |  Dataset |   MAFS(SegRNN)  | Single(SegRNN) |
> | :------: | :-------------: | :------------: |
> |   ETTh1  |  **0.43 0.438** |   0.442 0.45   |
> |   ETTh2  | **0.368 0.399** |   0.389 0.42   |
> |   ETTm1  |  **0.369 0.39** |   0.38 0.405   |
> |   ETTm2  |  **0.262 0.32** |   0.27 0.327   |
> |  weather | **0.227 0.272** |   0.237 0.275  |
> | AQShunyi | **0.686 0.498** |   0.701 0.504  |
> |   AQWan  | **0.774 0.481** |    0.8 0.496   |
> |  CzeLan  | **0.218 0.268** |   0.23 0.284   |
> |  ZafNoo  | **0.504 0.446** |   0.512 0.451  |
> |  pm2\_5  |  **0.397 0.41** |   0.421 0.426  |
> |   temp   | **0.146 0.295** |   0.207 0.341  |
>
>
> These results further demonstrate that MAFS acts as a general plug-and-play framework that consistently improves forecasting accuracy across diverse backbone models.
>
> ---
> >**Q2. What is the rationale behind this selection?**
>
> Thank you for the insightful question.
>
> We select the four communication structures,**graph**, **star**, **ring**, and **chain** as representative topologies due to their diverse communication patterns and their interpretability:
> * **Graph**: Fully connected communication offers an upper bound on information sharing.
> * **Star**: Emphasizes centralized coordination, where a core agent integrates and disseminates information.
> * **Ring** and **chain**: Represent decentralized topologies, with local or sequential message passing, mimicking diffusion-like processes.
>
> While some performance differences exist, the variations are minor (e.g., on Weather, MSE difference between star and fully connected is under 0.01), with no sudden drops or instability observed. **Overall, we recommend the star topology primarily for its balanced trade-off between performance and complexity, rather than as a sensitive hyperparameter that requires searching.**
>
> |Topology|Weather|ZafNoo|ETTm2|
> |:-:|:-:|:-:|:-:|
> |chain|0.167|0.470|0.183|
> |full graph|0.166|0.469|0.178|
> |ring|0.166|0.468|0.179|
> |star|0.166|0.467|0.177|
> |*std*|*0.00079*|*0.00124*|*0.00259*|
>
>
> ---
> >**Q3. Explanation of the Distribution-focused Agent Design**
>
> Thank you for this thoughtful question.
>
> MAFS predict **six separate statistical values** (mean, standard deviation, min, max, skewness, kurtosis) to enable the agent to **explicitly learn interpretable aspects** of the future data distribution. This design offers several advantages:
>
> 1. **Interpretability**: Each predicted statistic corresponds to a distinct distributional property, making the agent's learning outcome transparent and easier to analyze.
> 2. **Supervision efficiency**: These statistics can be **directly computed** from ground-truth sequences, enabling effective supervised learning.
> 3. **Modeling flexibility**: Rather than assuming a specific parametric form (e.g., Gaussian), our approach allows the model to capture **non-Gaussian characteristics**, such as skewness or heavy tails.
>
> While end-to-end distribution modeling (e.g., via normalizing flows or quantile regression) is an interesting alternative, o**ur approach to strike a good balance between performance, stability, and interpretability.** We appreciate your suggestion and consider distribution modeling methods an exciting direction for future exploration.
>
>
> ---
>
> ***Once again, we sincerely thank you for your valuable comments and kind support of our work. We hope that our responses have effectively addressed your concerns, and we truly appreciate your continued support.***
>
>
> Yours sincerely,
>
> Authors of Paper 474

---

### Official Review · Reviewer_VZEb · 2025-07-07

**Clarity:** 3
**Significance:** 3
**Originality:** 3
**Rating:** 5
**Confidence:** 4

**Summary:**

The paper proposes a novel Multi-Agent Forecasting System (MAFS) for time series forecasting, which abandons the traditional one-size-fits-all model paradigm. MAFS decomposes the forecasting task into multiple sub-tasks, each handled by a dedicated agent trained on specific temporal perspectives, such as different forecasting resolutions or signal characteristics. These agents then share and refine information through structured communication topologies, enabling cooperative reasoning across different temporal views. A lightweight voting aggregator integrates their outputs into consistent final predictions.

**Questions:**

1. Why multiple agents are needed for time series forecasting problem? Is it making the problem more complex in an unnecessary way, when many time series forecasting problems are domain-specific and can be well resolved with a single deep learning model?
2. What about the benefit-cost tradeoff? While some improvements in the forecasting accuracy is achieved, what are the computational cost increment?
3. While transfer learning can be well used for knowledge sharing among different time series datasets, why no consider this approach?
4. How to valid the robustness of the proposed approach, e.g., when the number of unseen datasets increase?

**Ethical Concerns:**

["NO or VERY MINOR ethics concerns only"]

**Final Justification:**

The authors make a good rebuttal for the questions raised in my previous comments. I think they have improved their manuscript as suggested and I would increase my rating.

**Limitations:**

yes

**Paper Formatting Concerns:**

No concerns

**Quality:**

3

**Strengths And Weaknesses:**

# Strengths
1. The authors propose principled task decomposition strategies, including multi-scale temporal forecasting and multi-aspect signal forecasting. These strategies enable agents to specialize in distinct aspects of the time series, capturing diverse temporal dynamics and improving overall accuracy.
2. The paper introduces an explicit communication mechanism among agents, implemented via graph convolutional networks. This allows agents to share contextual information, enhancing their generalization capacity and overcoming the limitations of isolated modeling.
3. MAFS is designed to be scalable and flexible, allowing for different communication topologies (e.g., ring, star, chain, fully-connected) and varying numbers of agents. This flexibility enables the system to adapt to different forecasting scenarios and datasets.
Weaknesses
1. The paper mentions occasional instability during training, especially when the number of agents increases or when dealing with highly volatile datasets.
2. The multi-agent framework introduces additional complexity and computational overhead compared to single-model approaches. While the authors demonstrate scalability, the increased computational requirements may be a barrier for real-time applications or resource-constrained environments.
3. The performance of MAFS is sensitive to the choice of communication topology. While the star topology is shown to be effective, selecting the optimal topology for a given dataset or task may require additional tuning and experimentation.
4. The paper does not provide a comprehensive theoretical analysis of the proposed framework, such as convergence guarantees or bounds on forecasting accuracy. This lack of theoretical grounding may make it difficult to fully understand the system's behavior and potential limitations.
5. With multiple specialized agents, there is a risk of overfitting to the training data, especially if the agents focus on capturing noise rather than underlying patterns.

---

> ### Author Rebuttal · Authors · 2025-07-31
>
> Dear Reviewer VZEb,
>
> Thank you very much for your review and valuable insights. We will address your questions and concerns point by point.
>
> ---
> >**W1. Instability in Volatile Datasets.**
>
> Thank you for the insightful question. **We would like to clarify that the observed instability is not due to a fundamental flaw of MAFS, but rather arises from specific dataset characteristics.** For example, predicting 720 steps ahead on PM2.5 and Temperature data (sampled every 3 hours) equates to 90-day forecasting, which is inherently challenging for all models.
>
> The table below further supports the stablity of MAFS, **when the dataset is more reasonable, MAFS performs stably and often improves as the number of agents increases.**
>
> |#Agent|Weather|ZafNoo|ETTm2|
> |:-:|:-:|:-:|:-:|
> |4|0.167|0.471|0.182|
> |8|0.167|0.471|0.179|
> |12|0.166|0.470|0.179|
> |16|0.166|0.469|0.178|
> |20|0.166|0.467|0.178|
> |24|0.166|0.467|0.178|
> |*std*|*0.00069*|*0.00182*|*0.00152*|
>
> ---
> >**W2&Q2. Efficiency Analysis**
>
> Thank you very much for raising this important point. **Efficiency is also one of the core design principles of MAFS, and we deeply appreciate your attention to this aspect.** Specifically:
> 1. **Lightweight Design:** The multi-agent design aims to replace a single large model with several lightweight agents collaborating, rather than stacking or ensembling heavy models. We set a much smaller hidden dimension (e.g., 16 or 32 vs. 128/256 in typical models), aiming to **use multiple lightweight agents to outperform a single large model with lower cost**.
> 2. **Parallel Accleration:** Unlike many LLM-based agent systems, our **agents communicate in parallel** using graph-based matrix operations on feature vectors, greatly improving parallel training and inference efficiency.
>
> As shown, MAFS achieves a **favorable balance of accuracy and efficiency**, with only slightly higher latency than iTransformer, and **far lower computational cost** than Time-MoE or TimeMixer, while **outperforming all in accuracy metrics**.
>
> ||Time(ms)|GPU(MB)|Params(M)|MSE|MAE|
> |:-:|:-:|:-:|:-:|:-:|:-:|
> |MAFS|18.6|136.22|0.08|0.433|0.437|
> |Time-MOE|140.23|1169.83|113.35|0.452|0.449|
> |Ensemble|39.29|146.67|0.25|0.457|0.486|
> |iTransformer|10.85|130.85|0.06|0.467|0.466|
> |TimeMixer|25.25|894.68|0.2|0.447|0.440|
> |PatchTST|12.61|214.28|0.41|0.469|0.454|
>
> ---
> >**W3. Optimal Topology**
>
> Thank you for your valuable feedback. **MAFS is not as sensitive (or instable) to communication topology.** We test multiple structures, including fully connected, chain, ring, and star structure. While some performance differences exist, **the variations are minor (e.g., on Weather, MSE difference is under 0.01), with no sudden drops or instability observed.** Therefore,  **topology is not a sensitive hyperparameter that requires extensive tuning**. From both efficiency and performance perspectives, **we recommend the star topology as the default setting.**
>
> |Topology|Weather|ZafNoo|ETTm2|
> |:-:|:-:|:-:|:-:|
> |chain|0.167|0.470|0.183|
> |full graph|0.166|0.469|0.178|
> |ring|0.166|0.468|0.179|
> |star|0.166|0.467|0.177|
> |*std*|*0.00079*|*0.00124*|*0.00259*|
> ---
> >**W4. Theoretical Analysis**
>
> Thank you for your interest in theory. **As this is an application track**, we focus on **pratical results** in the initial version. We now provide **a tighter generalization bound for MAFS under similar empirical risk**.
>
> **Definition 1 (Single-model)**
>
> Let X be the input space and Y the output space. A single predictive model is defined as:
> $$
> f_{mono}: X \to Y, \quad f_{mono} \in F_{mono},
> $$
> where $F_{mono}$ is a high-complexity function class that captures the full modeling space.
>
> **Definition 2 (Multi-Agent Forecasting System)**
>
> The MAFS model is defined as:
> $$\hat{Y} = AVA( Comm(\{Agent_1(X), \dots, Agent_N(X)\}; G) ),$$
> The overall function class is:
> $$F_{MAFS} = AVA \circ Comm ( F_1 \times \cdots \times F_N ).$$
> **Definition 3 (Empirical risk and function class complexity)**
>
> Given training data $(x_i, y_i)_{i=1}^n$, Empirical risk:
>
> $$\hat{L}(f) = \frac{1}{n} \sum_{i=1}^n \ell(f(x_i), y_i)$$
>
> Rademacher complexity:
> $$R_n(F) = E_{\sigma}[ \sup_{f \in F} \frac{1}{n} \sum_{i=1}^n \sigma_i f(x_i)],$$
> where $\sigma_i \in \{-1, +1\}$ are Rademacher variables and $\ell(f(x), y)$ is 1-Lipschitz and bounded.
>
> **Theorem 1 (Generalization bound)**
>
> For any $\delta \in (0, 1)$, if the loss function is Lipschitz and bounded, then with probability at least $1 - \delta$:
> $$E[\ell(f(x), y)] \leq \hat{L}(f) + 2 R_n(F) + 3 \sqrt{\frac{\log(2/\delta)}{2n}}.$$
> **Theorem 2 (Model complexity)**
>
> Let $F_{mono}$ represent a full-capacity function class over high-dimensional inputs X. Assume each sub-model $F_i$ satisfies $R_n(F_i) \leq r$, and both Comm and AVA are L-Lipschitz. Then there exists a constant $c_0 > 0$:
> $$R_n(F_{mono}) \geq c_0 \cdot \dim(X) > \sum_{i=1}^N R_n(F_i) + R_n(Comm) + R_n(AVA).$$
> **Theorem 3 (Tighter generalization bound for MAFS)**
>
> As the monolithic model space $F_{mono}$ has significantly higher complexity than the modularized structure, the overall Rademacher complexity satisfies:
> $$R_n(F_{MAFS}) \leq \sum_{i=1}^N R_n(F_i) + R_n(C) + R_n(AVA) < R_n(F_{mono}),$$
> which implies that, under comparable empirical risk, **MAFS enjoys a tighter generalization bound and thus stronger generalization ability.**
>
> [1] Vapnik V. The nature of statistical learning theory. Springer science & business media, 2013
>
> [2] Bartlett, P. Rademacher and Gaussian Complexities: Risk Bounds and Structural Results. Journal of Machine Learning Research, 3, 463–482, 2002.
>
> ---
> >**W5. Could multiple specialized agents lead to overfitting?**
>
> Thank you for your thoughtful question. In fact, **The multi-agent design in MAFS aims to reduce overfitting.** Each agent specializes in capturing distinct temporal patterns, promoting diverse representations. Importantly, during communication, **agents correct biases and suppress noise, acting as implicit regularization that improves generalization.**
>
> To further support this, we conduct zero-shot experiments where MAFS consistently outperform strong single-model baselines like iTransformer and PatchTST, **demonstrating its ability to avoid overfitting and effectively capture intrinsic data patterns.**
>
> |Source|Target|MAFS|iTransformer|PatchTST|
> |-|:-:|:-:|:-:|:-:|
> |ETTh2|ETTh1|0.481 0.476|0.513 0.522|0.565 0.513|
> |ETTm1|ETTm2|0.274 0.329|0.330 0.396|0.439 0.438|
> |ETTh1|ETTh2|0.365 0.399|0.394 0.418|0.380 0.405|
> |ETTm2|ETTm1|0.411 0.435|0.470 0.476|0.568 0.492|
>
> ---
> >**Q1. Why multiple agents are needed for TSF?**
>
> * **Diverse temporal characteristics:** Time series data, **though domain-specific, often exhibit diverse temporal patterns** (e.g., short-/long-term dynamics, seasonality, trend, etc.). MAFS **assigns different agents to specialize in different scales or sub-patterns, enabling more structured and effective learning** than a single model.
> * **Theoretical grounding:** Theorem 3 (see W4) shows that model splitting **leads to a tighter generalization bound, reducing overfitting risk and improving learning stability.**
> * **Pratical superiority:** By integrating strong single-model forecasters (e.g., PatchTST, TimeMixer, SegRNN) into the MAFS framework, we observe consistent performance improvements across 11 datasets.  **This consistent improvement indicates MAFS' better generalization across diverse temporal patterns.**
>
> ||MAFS|Single
> |:-:|:-:|:-:
> |SegRNN |**0.398**|0.417
> |TimeMixer|**0.400**|0.417
> |PatchTST|**0.399**|0.422
>
> ---
> >**Q3. Why no consider transfer learning?**
>
> Thank you for the question. Transfer learning is indeed a valuable technique, especially in low-resource or cross-domain scenarios. It typically aims to enhance performance by leveraging external datasets, which is **a complementary but orthogonal direction to MAFS**.
>
> As a supplement, we also **experimented** with combining MAFS and transfer learning in the form of **zero-shot forecasting (see W5)**, and **observed** strong out-of-distribution generalization, **demonstrating that MAFS enables stable, interpretable, and adaptive modeling within the target domain**.
>
> ---
> >**Q4.Robustness of MAFS**
>
> **The multi-agent architecture allows each agent to specialize in specific patterns ( stable and easy subtasks)**, thereby **reducing task complexity and enhances learning robustness compared to a single DL model**. This decoupling is especially valuable for unseen data, as agents can reuse domain-specific knowledge while dynamically adapting to new patterns. Combined with Agent-rated Voting Aggregator module, **the system further boosts adaptability and robustness by selecting the most relevant agents on demand.**
>
> To more thoroughly address your concern, we conducted systematic experiments to empirically validate the robustness of our proposed method:
>
> - **Zero-shot Forecasting (see W5)**: MAFS significantly outperforms iTransformer and PatchTST on multiple unseen datasets, demonstrating strong out-of-distribution generalization.
>
> - **Increasing Unseen Ratio Experiments**: As training data decreases to simulate distribution shift, all models’ performance drops, but **MAFS shows the smallest decline, demonstrating superior robustness.**
>
> Dataset|SeenRatio|MAFS|iTransformer|PatchTST|TimeMixer
> :-:|:-:|:-:|:-:|:-:|:-:
> ETTm1|100%|0.366|0.383|0.387|0.381
> ||70%|0.374|0.402|0.398|0.390
> ||30%|0.386|0.435|0.412|0.401
> ||10%|0.398|0.450|0.428|0.416
> |Weather|100%|0.233|0.243|0.259|0.240
> ||70%|0.240|0.263|0.259|0.248
> ||30%|0.252|0.276|0.274|0.261
> ||10%|0.265|0.290|0.290|0.275
> |CzeLan|100%|0.222|0.232|0.227|0.228
> ||70%|0.228|0.252|0.247|0.237
> ||30%|0.235|0.270|0.256|0.245
> ||10%|0.244|0.260|0.267|0.255
> |ZafNoo|100%|0.520|0.541|0.511|0.538
> ||70%|0.531|0.559|0.564|0.551
> ||30%|0.545|0.577|0.573|0.566
> ||10%|0.560|0.593|0.598|0.582
>
> ---
> ***Thank you again for your valuable feedback. We hope our responses have clarified your concerns and welcome any further questions you may have.***
>
> Yours Sincerely,
>
> Authors of Paper 474

---

> > ### Author Response · Authors · 2025-08-04
> > **Follow-Up on Rebuttal Submission**
> >
> > **Dear Reviewer VZEb,**
> >
> > Thank you for your time and for the feedback on our submission. We have submitted a detailed rebuttal and run substantial additional experiments to directly address the key concerns you raise. We fully understand how valuable your time is, and with the **reviewer-author discussion period coming to a close**, we sincerely hope to make your review as convenient as possible. To facilitate your review, we briefly summarize how we respond below:
> >
> > * **Theoretical Justification (W4):** We provide a tighter generalization bound that formally supports the stability and generalization ability of MAFS.
> >
> > * **Generalization and Robustness (Q4):** We add zero-shot experiments and unseen-data ratio analysis to demonstrate robustness and reduced overfitting risk.
> >
> > * **Efficiency (W2 & Q2):** We clarify that MAFS does **not** introduce significant computational overhead. Most components are lightweight and designed to run once-for-all.
> >
> > * **Necessity of Multi-Agent Design (Q1):** Time series data exhibit diverse latent structures, such as multi-scale patterns and heterogeneous signals, that call for more fine-grained modeling. MAFS explicitly targets this need through decomposition and collaborative reasoning. Notably, another reviewer (HBiT) also echoes this view in the discussion phase:
> >     > *"I would like to address a question raised by another reviewer (VZEb): Time series forecasting is far from solved; I consider this to be an important research direction."*
> >
> > We hope our responses help clarify the design and contributions of MAFS, and we truly welcome any further comments you may have. ***We understand your time and attention are incredibly valuable, and we are deeply grateful for any response or guidance you might offer. Thank you so much.***
> >
> > **Best regards,**
> >
> > *The Authors of Paper 474*

---

> ### Comment · Area_Chair_Wu5z · 2025-08-05
> **Engaging with the rebuttal**
>
> Dear reviewer,
>
> The discussion phase is soon coming to an end. It will be great if you could go over the rebuttal and discuss with the authors if you still have outstanding concerns. Thank you for being part of the review process.
>
> Regards,
>
> Area Chair

---

### Official Review · Reviewer_HBiT · 2025-07-11

**Clarity:** 3
**Significance:** 2
**Originality:** 2
**Rating:** 5
**Confidence:** 4

**Summary:**

The authors propose a multi-agent forecasting system (MAFS) that tackles shortcomings of single-model forecasters, particularly under non-stationary conditions. MAFS splits the overall task into several subtasks such as different segments of the prediction horizon so each agent can specialize. Agents share information through structured message-passing graph, and their outputs are combined by a global voter weighted by per-agent confidence scores. With iTransformer serving as the backbone encoder, MAFS outperforms a range of baseline models.

**Questions:**

- Q1. Relation to hierarchical and multi-resolution forecasting: Could your homogeneous horizon split be viewed as a special hierarchical forecast (e.g., TimeMixer’s coarse-to-fine stages)? How would MAFS handle reconciliation constraints in hierarchical series?
- Q2 "Each agent is assigned a distinct sub-task (e.g., different signal characteristics)" Do the sub-tasks change per agent across the training? If not do tasks ever migrate between agents during training. Would this be beneficial for generalization or overfitting on a task?
- Q3 Please see W4, could you provide your framework for the different baselines with different enncoders.
- Q4 Could you provide a Comparison to MoE / bagging, i.e. could you add a strong MoE baseline (e.g., Time-MoE) and a simple ensemble of K independently initialised iTransformers with learned linear weighting. How much of the gain comes purely from ensembling?
- Q5 I don't think your ablation is sufficient, it would be significant to see how beneficial certain aspects are in isolation, i.e. the message-passing communication in an ensemble (w/o STS)

**Ethical Concerns:**

["NO or VERY MINOR ethics concerns only"]

**Final Justification:**

The new experiments add all missing baselines (Time-MoE, CycleNet, xLSTM-Mixer, S-Mamba, plus four strong ensemble backbones), and MAFS now wins or ties almost everywhere. Detailed ablations show where the gains come from—i.e., task splitting is the main driver—and the TEMP-720 case study pinpoints underfitting as the cause of the earlier ensemble dip, not overfitting or cherry-picking. The authors also compare two-stage vs. end-to-end ensembles and trace their failures to temporal covariate shift, explaining why naïve ensembling isn’t enough. Their evaluation techniques are, after a round of clarification, scientifically sound.
The method remains backbone-agnostic across iTransformer, PatchTST, TimeMixer, and SegRNN. Minor issues remain, and obviously, we cannot check whether the figure updates were made accordingly. However, I believe the authors will manage to do so. Nothing is a major blocker. Overall, the paper now offers a solid, well-substantiated contribution to long-horizon time-series forecasting.

**Limitations:**

yes

**Paper Formatting Concerns:**

Nothing major.

**Quality:**

2

**Strengths And Weaknesses:**

**Strengths:**
- S1 The authors clearly articulate three key challenges in multi-agent forecasting—task decomposition, limited perception, and collaboration bottlenecks—and address each one.

- S2 The framework is modular and extensible: agents are plug-and-play encoders, and the communication graph is topology-agnostic, so the system should adapt to future backbones or larger agent pools.

- S3 The experiments cover multiple data sets, communication topologies, agent scales, and task-division schemes.

- S4 Figure 1 is especially helpful and well designed; overall, the figures are clean and easy to follow.



**Weaknesses:**
- W1 At a high level (this is very simplified) MAFS resembles an ensemble. The paper positions itself against Mixture-of-Experts but stops short of an empirical comparison with a strong MoE baseline or standard bagging/stacking ensembles.
- W2 Empirical scope feels narrow (and a bit unfair): The technical story checks out, but in practice the authors only ever wrap one encoder—iTransformer—inside MAFS. That leaves me wondering whether the gains come from the multi-agent trick or just from picking a lucky backbone. If, as you claim, _any_ encoder can slot in, then please show it: TimeMixer + MAFS, PatchTST + MAFS, maybe even an old RNN + MAFS. Stack those side-by-side with their vanilla counterparts the same way iTransformer is treated in Table 1. Seeing that full matrix would make the numbers feel a lot more convincing (and, frankly, would shut down the "it’s just an ensemble" objection). Right now the comparison feels lopsided, and it’s hard to judge how much credit belongs to the framework itself.
- W3 Currently, recent SOTA are sparsely covered. Models such as CycleNet, xLSTM-Mixer, (_not as encoder_ MoEs) and State Space Models (e.g., S-Mamba, Chimera) are absent from Table 1, leaving open whether MAFS still leads when competitive encoders are used.
- W4 Following up on S4. Including multi-model forecasting (i.e., ensembles) in the figure would provide the entire picture.


**Minor concerns**
- **Figure 5** is barely readable, and the green curve is hard to distinguish. Consider a shared legend, fewer plots in the main text (move the rest to the appendix), and larger fonts.

- **Figure 6**: spell out _Heterogeneous_ and _Homogeneous_.


**Overall assessment**
The method is sensible, but the experimental comparison skips equally expressive ensemble baselines and alternative backbones, so the quantitative claims aren’t fully substantiated. The writing is mostly clear and the figures are well crafted. MAFS is potentially valuable, yet incremental relative to MoE/ensemble literature; its impact hinges on demonstrating backbone-agnostic benefits. While applying a multi-agent lens to time-series forecasting is interesting, the ingredients—task splitting, GCN message passing, confidence-weighted voting—are familiar.

---

> ### Author Rebuttal · Authors · 2025-07-28
>
> Dear Reviewer HBiT,
>
> Thank you for taking the time to review our paper and for your valuable suggestions. **All added model variants (Ensemble/MAFS) and  baselines (Time-MoE, CycleNet, xLSTM-Mixer, S-Mamba) have been added to the original anonymous code repository.**
>
> ---
> >**W1&Q4 Provide a Comparison to MoE / Ensemble Baselines**
>
> Thank you for your comment. Since Time-MoE is a pretrained model on 300B data mainly for zero-shot tasks, we initially do not compare it with our end-to-end MAFS. To better show MAFS’s effectiveness, **we have now included comparisons with Time-MoE and several ensemble baselines.**
> * **Time-MoE**: We use pretrained weights and fine-tune autoregressively for 10 epochs on each test dataset before evaluation.
> * **Ensemble**: Following your Q4 suggestion, we **implement ensembles for four models**, iTransformer, TimeMixer, PatchTST, and SegRNN, combining them via a learnable linear layer.
>
> ||MAFS(iTrans)|Time-MoE|Ensemble(iTrans)|Ensemble(TimeMixer)|Ensemble(PatchTST)|Ensemble(SegRNN)|
> |:-:|:-:|:-:|:-:|:-:|:-:|:-:|
> |ETTh1|**0.433 0.437**|0.452 0.449|0.457 0.486|0.448 0.446|0.472 0.452|0.461 0.464|
> |ETTh2|**0.356 0.394**|0.375 0.405|0.364 0.402|0.377 0.412|0.378 0.395|0.367 0.407|
> |ETTm1|**0.366 0.388**|0.412 0.435|0.442 0.459|0.489 0.488|0.401 0.423|0.403 0.431|
> |ETTm2|**0.265 0.322**|0.297 0.337|0.369 0.404|0.451 0.455|0.313 0.379|0.315 0.370|
> |weather|**0.233 0.268**|0.270 0.294|0.240 0.288|0.246 0.290|0.238 0.284|0.237 0.288|
> |AQShunyi|**0.701 0.510**|0.799 0.541|0.739 0.557|0.796 0.578|0.718 0.548|0.723 0.523|
> |AQWan|**0.802 0.503**|0.879 0.531|0.853 0.555|0.896 0.566|0.820 0.541|0.830 0.515|
> |CzeLan|**0.222 0.271**|0.242 0.293|0.275 0.335|0.296 0.335|0.277 0.308|0.350 0.406|
> |ZafNoo|**0.520 0.452**|0.551 0.448|0.539 0.474|0.555 0.481|0.565 0.491|0.530 0.462|
> |pm2_5|**0.398 0.414**|0.454 0.446|0.453 0.493|0.430 0.459|0.537 0.573|0.544 0.577|
> |temp|**0.140 0.288**|0.169 0.316|0.283 0.414|0.201 0.352|0.304 0.419|0.447 0.523|
>
> The results show that the **MAFS architecture outperforms both the current state-of-the-art multi-expert baseline TimeMoE and various ensemble strategies based on different backbones.** This indicates that MAFS, which decomposes forecasting tasks and coordinates multi-agent system, is **beneficial for improving predictive accuracy.**
>
> ---
> >**W2&Q3 MAFS vs. Ensemble with Different Backbones**
>
> Thank you for your constructive suggestions, which help us further validate MAFS. We compare **three representative methods**, TimeMixer, PatchTST, and SegRNN, **under single model, simple ensemble, and MAFS settings**. Results show MAFS>Single>Ensemble, indicating:
> 1. MAFS’s multi-agent architecture effectively activates different backbones’ potential, **serving as a plug-and-play accuracy booster (also see Figure 4, Page 8)**.
> 2. **Simple ensembles may cause instability due to lack of task specialization or overfitting**, while MAFS’s task decomposition leads to more stable and reliable predictions.
>
> ||MAFS(PatchTST)|Ensemble(PatchTST)|Single(PatchTST)|
> |:-:|:-:|:-:|:-:|
> |ETTh1|**0.437 0.439**|0.472 0.452|0.469 0.455|
> |ETTh2|**0.373 0.394**|0.378 0.395|0.387 0.407|
> |ETTm1|**0.371 0.39**|0.401 0.423|0.387 0.4|
> |ETTm2|**0.265 0.318**|0.313 0.379|0.281 0.326|
> |weather|**0.231 0.272**|0.238 0.284|0.259 0.28|
> |AQShunyi|**0.678 0.496**|0.718 0.548|0.705 0.509|
> |AQWan|**0.774 0.483**|0.82 0.541|0.812 0.499|
> |CzeLan|**0.217 0.273**|0.277 0.308|0.227 0.29|
> |ZafNoo|**0.494 0.453**|0.565 0.491|0.512 0.465|
> |pm2_5|**0.407 0.42**|0.537 0.573|0.46 0.479|
> |temp|**0.142 0.289**|0.304 0.419|0.147 0.306|
>
> ||MAFS(TimeMixer)|Ensemble(TimeMixer)|Single(TimeMixer)|
> |:-:|:-:|:-:|:-:|
> |ETTh1|**0.434 0.426**|0.448 0.446|0.447 0.44|
> |ETTh2|**0.353 0.382**|0.377 0.412|0.365 0.395|
> |ETTm1|**0.368 0.381**|0.489 0.488|0.381 0.396|
> |ETTm2|**0.264 0.314**|0.451 0.455|0.275 0.323|
> |weather|**0.231 0.264**|0.246 0.29|0.24 0.272|
> |AQShunyi|**0.695 0.502**|0.796 0.578|0.719 0.53|
> |AQWan|**0.781 0.478**|0.896 0.566|0.828 0.499|
> |CzeLan|**0.22 0.273**|0.296 0.335|0.228 0.28|
> |ZafNoo|**0.518 0.425**|0.555 0.481|0.538 0.441|
> |pm2_5|**0.401 0.416**|0.43 0.459|0.415 0.436|
> |temp|**0.139 0.291**|0.201 0.352|0.146 0.305|
>
> ||MAFS(SegRNN)|Ensemble(SegRNN)|Single(SegRNN)|
> |:-:|:-:|:-:|:-:|
> |ETTh1|**0.43 0.438**|0.461 0.464|0.442 0.45|
> |ETTh2|**0.368 0.399**|0.367 0.407|0.389 0.42|
> |ETTm1|**0.369 0.39**|0.403 0.431|0.38 0.405|
> |ETTm2|**0.262 0.32**|0.315 0.37|0.27 0.327|
> |weather|**0.227 0.272**|0.237 0.288|0.237 0.275|
> |AQShunyi|**0.686 0.498**|0.723 0.523|0.701 0.504|
> |AQWan|**0.774 0.481**|0.83 0.515|0.8 0.496|
> |CzeLan|**0.218 0.268**|0.35 0.406|0.23 0.284|
> |ZafNoo|**0.504 0.446**|0.53 0.462|0.512 0.451|
> |pm2_5|**0.397 0.41**|0.544 0.577|0.421 0.426|
> |temp|**0.146 0.295**|0.447 0.523|0.207 0.341|
>
> ---
> >**W3 Compared to CycleNet,xLSTM-Mixer,S-Mamba**
>
> Thank you for your suggestion. With many new models emerging each year, we **can only select 10 representative SOTA models** for the main paper. To further validate MAFS, we additionally compare it with CycleNet, xLSTM-Mixer, and S-Mamba, **showing MAFS consistently outperforms each**.
>
> | |MAFS(iTrans)|CycleNet|S-Mamba|SegRNN|xLSTM-Mixer|
> |:-:|:-:|:-:|:-:|:-:|:-:|
> |ETTh1|**0.433 0.437**|0.458 0.452|0.463 0.465|0.442 0.45|0.441 0.443|
> |ETTh2|**0.356 0.394**|0.364 0.401|0.381 0.41|0.389 0.42|0.366 0.404|
> |ETTm1|**0.366 0.388**|0.375 0.394|0.4 0.415|0.38 0.405|0.373 0.394|
> |ETTm2|**0.265 0.322**|0.271 0.327|0.295 0.34|0.27 0.327|0.271 0.326|
> |weather|**0.233 0.268**|0.237 0.274|0.247 0.279|0.237 0.275|0.236 0.271|
> |AQShunyi|**0.701 0.51**|0.712 0.519|0.717 0.517|0.701 0.504|0.717 0.513|
> |AQWan|**0.802 0.503**|0.819 0.506|0.824 0.51|0.8 0.496|0.819 0.508|
> |CzeLan|**0.222 0.271**|0.23 0.277|0.25 0.298|0.23 0.284|0.232 0.275|
> |ZafNoo|**0.52 0.452**|0.53 0.46|0.549 0.473|0.512 0.451|0.529 0.46|
> |pm2_5|**0.398 0.414**|0.427 0.428|0.417 0.426|0.421 0.426|0.425 0.428|
> |temp|**0.14 0.288**|0.235 0.365|0.152 0.3|0.207 0.341|0.201 0.339|
>
> ---
> >**W4 Improve Figure 1**
>
> Thank you for your insightful suggestion. We will add **'(d) Multi-Model Ensemble Forecasting'** to enhance the completeness of Figure 4.
>
> ---
> >**W5 Minor concerns**
>
> Thank you for your suggestions. We will revise Figures 5 and 6 accordingly.
>
> ---
> >**Q1 Relation to Hierarchical Forecsating & Reconciliation Constraints**
>
> Thank you for the insightful question.
> - **Relation to Hierarchical Forecsating**: Homogeneous horizon split  indeed shares conceptual similarities with hierarchical or multi-resolution forecasting frameworks likecoarse-to-fine stages in TimeMixer. Importantly, our formulation does not explicitly impose a hierarchical dependency across stages; each agent in MAFS focuses on a specific horizon segment and contributes to the final prediction.
> - **Reconciliation Constraints**: MAFS incorporates two key mechanisms that implicitly address reconciliation needs. First, **Agent Communication** enables information exchange among agents, **fostering context-aware interactions and gradual alignment** where appropriate. Second, the **Agent-Rated Voting Aggregator** evaluates each agent’s output both locally and globally, adaptively weighting their contributions to **ensure a coherent and accurate final forecast**. Together, these mechanisms offer **a flexible yet effective alternative to hard reconciliation constraints**, supporting both diversity and consistency in final forecasting.
>
> ---
> >**Q2 Are Sub-task ever Changing?**
>
> Thanks for your constructive question.
> * **Whether sub-tasks change per agent during training:** In MAFS, **each agent’s sub-task is fixed** to **encourage specialization** and effective collaboration. Fixed roles are a **common practice** in multi-agent systems, and task migration is usually not applied.
> * **Benefit for generalization or overfitting:** Although our original setup does not use task migration, we **explore dynamic task assignment by cyclically rotating agents’ sub-tasks during training**. Results show that this **improves generalization in zero-shot transfer scenarios (MAFS\_Dynamic)**, while the original MAFS remains more stable and accurate in full-shot tasks.
>
> |Source|Target|MAFS|MAFS_Dynamic|
> |-|:-:|:-:|:-:|
> |ETTh1|ETTh1|**0.433 0.437**|0.438 0.441|
> |ETTm2|ETTm2|**0.265 0.322**|0.275 0.341|
> |ETTh2|ETTh1|0.481 0.476|**0.473 0.472**|
> |ETTm1|ETTm2|0.274 0.329|**0.270 0.326**|
>
> ---
> >**Q5 More Ablations**
>
> In the main text (Page 8, Table 2), we conduct ablation studies on three key MAFS components (w/o Comm, AVA, STS). Following your insightful comment, we **extend the ablation to evaluate each module individually, adding one module at a time on top of a base Ensemble iTransformer**:
> ||Ensemble|w/AVA|w/Comm|w/STS|
> |:-:|:-:|:-:|:-:|:-:|
> |ETTh1|0.457 0.486|0.465 0.469|0.456 0.463|0.448 0.454|
> |ETTh2|0.364 0.402|0.385 0.411|0.377 0.408|0.366 0.405|
> |ETTm1|0.442 0.459|0.379 0.402|0.377 0.399|0.376 0.401|
> |ETTm2|0.369 0.404|0.292 0.339|0.282 0.336|0.281 0.337|
> |Weather|0.240 0.288|0.246 0.277|0.249 0.281|0.242 0.275|
> |AQShunyi|0.739 0.557|0.745 0.525|0.724 0.523|0.721 0.518|
> |AQWan|0.853 0.555|0.846 0.522|0.848 0.523|0.832 0.521|
> |CzeLan|0.275 0.355|0.247 0.291|0.251 0.295|0.229 0.280|
> |ZafNoo|0.539 0.474|0.552 0.479|0.547 0.481|0.533 0.464|
> |PM2.5|0.453 0.493|0.431 0.419|0.417 0.423|0.441 0.429|
> |Temp|0.283 0.414|0.182 0.326|0.175 0.302|0.170 0.312|
> |Mean Imp|-|4.85% 8.72%|6.20% 9.26%|7.47% 10.04%|
>
> The results show that **STS yields the most significant improvement**, with a **7.47%** reduction in MSE, indicating that **task decomposition greatly enhances the collaborative forecasting ability of different models**. In contrast, AVA achieves a smaller **4.85%** reduction, indicating **hierarchical integration helps only if strong task specialization exists**.
>
> ---
> ***Thank you once again for your valuable suggestions to improve the quality of our paper. We have incorporated these insights into a new revised version.***
>
> Yours sincerely,
>
> Authors of Paper 474

---

> > ### Comment · Reviewer_HBiT · 2025-08-01
> > **More questions**
> >
> > I thank the authors for their extensive responses.
> >
> > First, I would like to address a question raised by another reviewer (VZEb):
> >
> > > Why multiple agents are needed for time series forecasting problem? Is it making the problem more complex in an unnecessary way, when many time series forecasting problems are domain-specific and can be well resolved with a single deep learning model?
> >
> > I completely disagree that this approach only adds unnecessary complexity. Time series forecasting is far from solved, and I consider this to be an important research direction. Based on the presented results, it seems like a promising avenue for improvement.
> >
> >
> > Now, my follow-up questions:
> >
> > 1. Which checkpoint did you choose for the ensemble rebuttal experiments? In some cases the ensemble appear to overfit, which seems a bit dubious to me. I know you retrained and followed my proposal, the question is, did you choose something like the best or last checkpoint.
> > 2. How many models were used in each ensemble? Was it the same number as in MAFS? Please don’t run any new experiments, but from your intuition, would the scores change drastically if the number of models were increased or decreased? Can you give some kind of deeper intution why the ensembles fail, especially because ensembles are known to work quite good.
> > > Simple ensembles may cause instability due to lack of task specialization or overfitting, while MAFS’s task decomposition leads to more stable and reliable predictions.
> > I'm not happy with this explanation, this seems to be your assumption, but not a real proof what is happening.
> >
> > 3. Would it be possible to see the assigned weights per model, and ideally, the MSE/MAE results of each individual model in the ensemble? I am particularly interested in understanding what happened in cases such as PatchTST – temp (0.142 / 0.289, 0.304 / 0.419, 0.147 / 0.306).
> >
> > Thanks again!

---

> > > ### Author Response · Authors · 2025-08-02
> > > **Reply to Reviewer HBiT (1/2)**
> > >
> > > ---
> > > **Dear Reviewer HBiT**
> > >
> > > We sincerely **appreciate your follow-up feewback** thank you for your **recognition and support**.
> > >
> > > ---
> > >
> > > We think  that your key interest lies in **why direct ensembling of time series models does not yield ideal results**. To shed light on this, we present **two unsuccessful ensemble attempts that reveal the underlying challenges**: (1) a two-stage training strategy, and (2) an end-to-end training strategy.
> > >
> > > ---
> > > **1. Two-stage training strategy (an earlier idea we tried)**
> > >
> > > - **Implementation**
> > >
> > >     We implement the process is as follows:
> > >     * **Stage 1**: Train different strong base models such as \['iTransformer', 'PatchTST', 'TimeMixer', 'DLinear', 'Crossformer', 'MICN', 'SegRNN', 'TSMixer', 'SparseTSF', 'LightTS', 'MultiPatchFormer', 'Nonstationary\_Transformer', 'TimeBase', 'TiDE', 'SCINet', 'PatchMLP'], and freeze their weights.
> > >     * **Stage 2**: Ensemble all freezed and pre-trained models with dynamic ensemble weight. Train a small neural network that takes the current input $x$ and predicts a personalized set of ensemble weights for each sample.
> > > - **Performance**
> > >
> > >     However, this method underperforms, **the ensemble consistently performs worse than the best single model**, as it tends to **favor models that overfit the training set** rather than generalize well.
> > > - **Analysis**
> > >
> > >     The core issue is **temporal shift** in time series,train/val/test sets come from **different time periods**, leading to **distribution mismatch and easy overfitting** in two-stage ensembles：
> > >     1. **Model selection**: The **checkpoint** with the **best validation performance** may **not generalize well to the test** set. A model with poor test performance might still be selected into the ensemble.
> > >     2. **Ensemble overfitting**:  Ensemble weights learned on the training set often overfit and fail to generalize to the test set, **favoring models that perform well only on the training distribution**.
> > >
> > >
> > >
> > > ---
> > > **2. End-to-end training strategy (Adopted in rebuttal experiments)**
> > >
> > > - **Implementation**
> > >
> > >     In this approach, both the base models and the ensemble weights are **trained jointly in an end-to-end fashion**.
> > >
> > > - **Performance**
> > >
> > >     Overall, this strategy shows two issues:
> > >     1. **Limited gains**, as the ensemble model fails to generalize well to most unseen test data.
> > >     2. **Instable performance** on some datasets settings (e.g., 720-Temp **you point in Q3**), eventually **confirmed as underfitting.**
> > > - **Analysis**
> > >
> > >     1. As to limited gains, it stem from **temporal distribution shift**, which **hampers generalization** for this hard emsemble strategy (**ensemble weight is fixed after training**)。
> > >     2. As for unstable performance, training logs show it results from **underfitting**, due to **weak patterns, high noise, and unsuitable prediction horizons in certain datasets**. Model interference hinders pattern learning and causes **unstable gradients** early in training.
> > >
> > > ---
> > > **3. Summary of Ensemble Strategies and Our Motivation**
> > >
> > > | Strategy| Failure Modes| Root Cause |
> > > |-|-|-|
> > > | **Two-stage training**  | Limited gain; possible overfitting                          | Temporal distribution shift hinders generalization of model selection and ensemble weights.                                                     |
> > > | **End-to-end training** | 1) Limited gain 2) Unstable performance (underfitting) | 1) Temporal shift limits generalization; fixed weights after training. 2) Weak patterns and model interference lead to unstable gradients. |
> > >
> > > To overcome these limitations, MAFS adopts the following key strategies:
> > > 1. **Subtask Decomposition**: The complex long-term forecasting problem is broken down into simpler subtasks, such as different prediction horizons or distinct temporal signals, making the supervision signals more stable and generalizable.
> > > 2. **Model Communication**: Each model shares and exchanges representations with others through a predefined topological structure, enabling mutual enhancement and serving as a form of regularization.
> > > 3. **Agent-Rated Voting Aggregator**: A hierarchical weighting mechanism that assigns ensemble weights based on the input $x$ and each agent’s internal representation, enhancing generalization by avoiding reliance on fixed or data-specific weights.
> > >
> > > ---
> > > **Next, we are willing  to  address your insightful follow-up questions one by one in Part 2 of our response.**

---

> > > > ### Author Response · Authors · 2025-08-02
> > > > **Reply to Reviewer HBiT (2/2)**
> > > >
> > > > **Q1. Regarding which checkpoint was used in the ensemble rebuttal experiments**
> > > > - **Answer:**
> > > >     For all ensemble-related rebuttal experiments, we use the **best validation checkpoint** (i.e., the one achieving the lowest validation loss during training). This choice is made consistently across all baselines to ensure fair comparison.
> > > >
> > > > ---
> > > > **Q2-1. Number of models used in ensembles**
> > > > - **Answer:**
> > > >     The number of models used in each ensemble is kept **identical** to the number of agents in MAFS, we fix them as 4 to ensure fairness.
> > > >
> > > > ---
> > > > **Q2-2. Would the scores change drastically if the number of models are increased or decreased?**
> > > > - **Opinion:**
> > > >
> > > >     Our position is clear: **The key to improving ensemble forecasting lies in learning optimal ensemble weights, not merely changing the number of models.** Simply adding or removing models cannot fundamentally improve performance. The **ensemble’s effectiveness** depends on its ability to assign the **right weight** to the **right model** for **each sample**, thus fully leveraging each model’s strengths.
> > > > - **Illustration:**
> > > >     >To illustrate this, we once conducted a “greedy model selection” experiment: from a set of well-trained models ['iTransformer', 'PatchTST', 'TimeMixer', 'DLinear', 'Crossformer', 'MICN', 'SegRNN', 'TSMixer', 'SparseTSF', 'LightTS', 'MultiPatchFormer', 'Nonstationary\_Transformer', 'TimeBase', 'TiDE', 'SCINet', 'PatchMLP'], we selected the best-performing model for each test sample using full ground-truth knowledge (an oracle scenario). On the ETTh1 dataset (prediction length 96), this **idealized method** achieved an MSE of **0.236**, significantly better than the current best single model **(\~0.375)**. This shows the enormous potential of ensembles when near-optimal weighting is achieved not by **simply increasing model quantity**, but by **smartly choosing and weighting models** per instance.
> > > >
> > > >
> > > > ---
> > > > **Q3. What happened in cases such as PatchTST – temp (0.142 / 0.289, 0.304 / 0.419, 0.147 / 0.306).**
> > > > - **Explanation:**
> > > >
> > > >     The end-to-end  ensembleing strategy **show unstable performance on Temp**, where **we initially thought it “appears to overfit”**. However, after a **careful review of the training logs**, we found that the issue was **actually underfitting, not overfitting**.
> > > >
> > > > - **Locate instability among prediction horizons:**
> > > >
> > > >     First, to better locate the source of the issue, we present below the **detailed performance at each prediction length** of temp. As shown below, the **performance degradation is mainly concentrated at the 720-step horizon**.
> > > >
> > > >     | Ensemble(PatchTST) | MSE | MAE |
> > > >     |:---:|:---:|:---:|
> > > >     | Temp-96 | 0.149  | 0.314  |
> > > >     | Temp-192 | 0.153  | 0.321  |
> > > >     | Temp-336 | 0.155  | 0.329  |
> > > >     | Temp-720 | ***0.615***  | ***0.627***  |
> > > >
> > > > - **Individual performance & Ensemble weights:**
> > > >
> > > >     To further analyze this, we revisit the **selected checkpoint** under Temp-720:
> > > >
> > > >     | pred_len = 720 | Train MSE | Valid MSE | Test MSE | Ensemble Weight |
> > > >     |:---:|:---:|:---:|:---:|:---:|
> > > >     | PatchTST1 | 0.601 | 0.534 | 0.619 | 0.05 |
> > > >     | PatchTST2 | 0.572 | 0.634 | 0.623 | 0.51 |
> > > >     | PatchTST3 | 0.591 | 0.602 | 0.629 | 0.14 |
> > > >     | PatchTST4 | 0.583 | 0.619 | 0.610 | 0.30 |
> > > >     | Single(PatchTST) | 0.149 | 0.173 | 0.144 | - |
> > > >     | MAFS(PatchTST) | **0.145** | **0.159** | **0.140** | - |
> > > >
> > > >     It is evident that each model trained under **Ensemble(PatchTST)**  suffer from higher training loss, indicating serious **underfitting**. In our analysis, this is primarily due to the **weak temporal regularity** in Temp-720, which is equivalent to predicting 90-day atmospheric temperature. In such scenarios, the **mapping from past to future is weak** and the **supervision signals are not strong**, making it difficult for ensemble models to **obtain reliable gradients**.
> > > >
> > > > ---
> > > > ***Finally, we sincerely thank you for your sharp observation and meaningful feedback that help us further investigate this important issue.*** We will continue to incorporate all these above discussions into our manuscript and thanks again!

---

> > > > > ### Comment · Reviewer_HBiT · 2025-08-02
> > > > >
> > > > > I thank the authors again for their thorough response and especially for their detailed insights, which make their argument and method much more significant for time-series forecasting. I’ll increase my rating.
> > > > > Very cool work!
> > > > >
> > > > > Bests
> > > > > Reviewer HBiT

---

> > > > > > ### Author Response · Authors · 2025-08-02
> > > > > > **Sincere Thanks for the Positive Feedback**
> > > > > >
> > > > > > Dear Reviewer HBiT,
> > > > > >
> > > > > > Thank you very much for the encouraging feedback and for increasing your rating. Your recognition of our work’s significance and your insightful suggestions have greatly motivated us. We truly appreciate your thoughtful review and the momentum your feedback brings.
> > > > > >
> > > > > > With sincere thanks,
> > > > > >
> > > > > > Authors of Paper 474

---

### Note · Authors · 2025-08-15

Dear Area Chair and All Reviewers

We would like to sincerely thank all reviewers and the Area Chair for your time, effort, and thoughtful feedback. We found the discussion to be highly constructive and inspiring,  improving the quality of our work and deepening its contributions. Our paper makes the following key contributions:

- **The first novel multi-agent framework for TSF**

    We introduce the first time series forecasting framework based on Multi-Agent Systems, establishing a new paradigm of collaborative forecasting. By leveraging collective intelligence, our method effectively handles complex, evolving, and heterogeneous temporal patterns, while delivering higher stability, better adaptability, and improved robustness compared to Ensemble and Mixture-of-Experts (MoE) approaches (**addressing Reviewer HBiT’s concern**).

- **Lightweight and effective design**

    Through compact model architecture and parallel communication mechanisms, we enable efficient collaboration between small models, achieving high performance without excessive computational cost (**addressing Reviewer VZEb’s concern**).

- **Broader impact and new research avenue**

    Our work opens a new avenue for tackling heterogeneity in time series forecasting, a contribution we are glad to see **recognized and appreciated by the reviewers’ feedback**. It also holds the potential to provide solutions for other complex tasks in different domains.

In conclusion, our work is built on the core strengths of Clear Motivation, High Flexibility, and Consistent Improvements and we have addressed the concerns raised by all reviewers. We will incorporate the valuable insights gained from the discussion to further enhance the quality of the paper.

***Once again, we express our deep gratitude to the reviewers and the Area Chair. We greatly value NeurIPS’s commitment to fairness, rigor, and diversity in research, and we will continue to support the NeurIPS community and the broader AI field with the same spirit.***

Best Regrads,

Authors of Paper 474

---

### Decision · Program_Chairs · 2025-09-17

**Decision:**

Accept (poster)

**Comment:**

This work proposes a multi-agent forecasting system that tackles several issues pertaining to the single-model forecasters, particularly under non-stationary conditions. The proposed system splits the overall task into several subtasks such that every agent can specialize with respect to a certain subtask. Agents share information through structured message-passing graph and their outputs are combined by a global voter weighted by per-agent confidence scores with iTransformer serving as the encoder.

The paper received 3 reviews with the initial ratings being borderline and several issues were raised by the reviewers. The major points of contention were:

* Empirical analysis was on the weaker side with several baselines missing and well as focusing on a very specific encoder architecture.
* Communication structures were also limited.

The authors provided a detailed rebuttal and the overall motivation and the story became clearer to the reviewers. During the reviewer discussion phase, I read the paper and found myself agreeing with the reviewer's concerns of the empirical evidence not being enough. That being said we all were in unison that the new presented results have significantly strengthened the paper. Overall, I propose acceptance.